



# Improved Soil Evaporation Remote Sensing Retrieval Algorithms and Associated Uncertainty Analysis on the Tibetan Plateau

Jin Feng[1,2,3], Ke Zhang[1,2,3,4*], Huijie Zhan[1], Lijun Chao[1,2,3]

[1] State Key Laboratory of Hydrology-Water Resources and Hydraulic Engineering, and College of Hydrology and Water Resources, Hohai University, Nanjing, Jiangsu, 210098, China

[2] Yangtze Institute for Conservation and Development, Hohai University, Nanjing, Jiangsu, 210098, China

[3] CMA-HHU Joint Laboratory for Hydro-Meteorological Studies, Hohai University, Nanjing, Jiangsu, 210098, China

[4] Key Laboratory of Water Big Data Technology of Ministry of Water Resources, Hohai University, Nanjing, Jiangsu, 210098, China

*Corresponding to: Ke Zhang (kzhang@hhu.edu.cn)*

**Abstract.** Actual evapotranspiration (ET) is the key link between water and energy cycles. However, accurate evaporation estimation in alpine barren areas remains understudied. In this study, we aimed to improve the satellite-
driven Process-based Land Surface ET/Heat fluxes algorithm (P-LSH) by introducing two frameworks for quantifying moisture constraints to ET, and to test the applicability of satellite soil moisture and precipitation data for improving ET retrieval. As a result, it formed two improved P-LSH algorithms. The first framework normalizes the surface soil moisture to represent moisture stress, while the second framework takes the ratio of cumulative precipitation to cumulative equilibrium evaporation to quantify soil water stress. We systematically assessed the
performances of the two improved P-LSH algorithms and six existing remote sensing ET retrieval algorithms on two barrens-dominated basins of the Tibetan Plateau using reconstructed ET estimates derived from the terrestrial water balance method as a benchmark. The two frameworks largely improved the performance of the P-LSH algorithm and showed better performance in both basins (root mean square error (RMSE) = 7.36 and 7.76 mm month$^{-1}$; $R^2$ = 0.86 and 0.87), resulting in a higher simulation accuracy than all six existing algorithms. We used five soil
moisture and five precipitation datasets to investigate the impact of moisture constraint uncertainty on the improved P-LSH algorithm. The ET estimates of the improved P-LSH algorithm, driven by the GLDAS_Noah soil moisture, performed best compared with those driven by other soil moisture and precipitation datasets, while ET estimates driven by various precipitation datasets generally showed a high and stable accuracy. These results suggest that high-quality soil moisture can optimally express moisture supply to ET, and that more accessible precipitation data
can serve as a substitute for soil moisture as an indicator of moisture status for its robust performance in barren evaporation.

## 1 Introduction

As a key link between the water and energy cycles, actual evapotranspiration (ET) is critical for assessing regional water and energy balances. Oki and Kanae (2006) reported that approximately 60% of precipitation returns to the



atmosphere in the form of ET, whereas the proportion can reach more than 90% in arid and semi-arid regions (Glenn et al., 2007; Morillas et al., 2013). Hence, accurate ET estimation is extremely important for irrigation planning, watershed management, and meteorology and climate change studies in arid and semi-arid regions.

Satellite remote sensing is an important means of estimating regional and global ET. A series of ET estimation algorithms have been developed over the past decade, including remote-sensing-based physical models, process-
based land surface models, and vegetation-index-based empirical algorithms. In remote-sensing-based physical models, the Penman-Monteith (PM) method (Monteith, 1965; Cleugh et al., 2007; Mu et al., 2011; Zhang et al., 2010a) and Priestley-Taylor (PT) method (Fisher et al., 2008; Martens et al., 2017; Priestley and Taylor, 1972; Yao et al., 2013) are the main representative methods for estimating ET. Several studies have combined these two methods to calculate canopy transpiration and soil evaporation (Leuning et al., 2008; Wang et al., 2018; Zhang et al.,
2019). The PT equation simplifies the PM equation and avoids the difficulty of quantifying aerodynamic and surface conductance. However, the PT equation simplifies the physical process, leading to a weaker physical basis than that of the PM equation. Land surface models reflect interactions and feedback between physical, biological, and biogeochemical processes in a predictive manner (Jiménez et al., 2011). These methods do not require remote sensing data; however, different parameterization schemes in land surface models for various physiological
processes lead to considerable uncertainty in ET estimation (Famiglietti and Wood, 1991; Pan et al., 2020). In addition, ET has a close relationship with the ecophysiological processes that can be represented by satellite spectral products such as the normalized difference vegetation index (NDVI), leaf area index (LAI), and land surface temperature (LST); as a result, a number of vegetation-index-based empirical algorithms have been developed (Wang et al., 2006; Glenn et al., 2010). Subsequent developments in machine learning have attracted further
attention in ET estimation because of their advantages in capturing the complex and nonlinear relationship between ET and its controlling environmental factors (Abdullah et al., 2015; Bai et al., 2021; Jung et al., 2010).

Although considerable effort has been made to estimate ET using the above methods, there are still significant uncertainties in quantifying the temporal and spatial characteristics and components of regional ET, especially in arid and semi-arid regions (Miralles et al., 2016; Pan et al., 2020). ET in these regions is dominated by water supply
and climatic water deficits, whereas in humid regions it is dominated by available energy (Vinukollu et al., 2011; Zhang et al., 2016a). It is worth studying how to accurately reflect the influences of water supply and climatic water deficits. In remote-sensing-based physical models, both the PM and PT equations use the moisture constraint $f$ to downscale the equilibrium (i.e., potential) evaporation at the soil surface to actual soil evaporation. Based on the hypothesis that surface moisture status is related to the adjacent atmospheric humidity (Bouchet, 1963), Fisher et al.
(2008) used relative humidity (RH) and vapor pressure deficit (VPD) to reflect soil moisture supply and atmospheric water deficit and applied this method to a wide variety of ecosystems, vegetation types, footprints, and climatic regimes. Zhang et al. (2010b) selected the cumulative precipitation and cumulative equilibrium evaporation rates over the past 32 days to estimate $f$. These results were verified in flux towers representing forests, grasslands, and savannas in Australia. Subsequently, a continuous ET dataset, including each component, was generated based on
this method (Zhang et al., 2016b; Zhang et al., 2019). Morillas et al. (2013) improved the method proposed by Zhang et al. (2010b) by adding a soil drying simulation factor after rainfall events and compared the uncertainties



between three different methods in semi-arid and sub-humid flux towers in the Mediterranean. Miralles et al. (2011) also identified environmental factors that constrain potential evaporation by the moisture constraint $f$, parameterized for tall canopies, short vegetation, and barren areas. For barrens with sparse vegetation, the $f$ estimates are based

only on surface soil moisture (θ) conditions (Miralles et al., 2011; Martens et al., 2017), and soil moisture is normalized by the wilting point and critical moisture level, with an exponential (subsequently simplified to linear) form to estimate $f$. However, this method relies heavily on soil properties. Yao et al. (2013) incorporated diurnal temperature changes into Apparent Thermal Inertia (ATI) estimation to calculate the moisture constraint $f$; this method was then compared with the relative extractable water (REW) of 16 flux towers in China and showed good

agreement. García et al. (2013) also expressed the moisture constraint $f$ using ATI, which was calculated using LST and albedo from the Meteosate Second Generation-Spinning Enhanced Visible and InfraRed Imager (MSG-SEVIRI) satellite. Their results showed that ET estimates derived from both towers and satellites performed better than the two-source model or the Penman-Monteith-Leuning model in the African Sahelian savanna and Mediterranean grasslands. However, this ATI-based method requires fine spatial and temporal resolutions of LST. Brust et al.

(2021) calculated REW as moisture control directly, using soil moisture data from the NASA Soil Moisture Active Passive (SMAP) mission. Their results showed that the accuracy of the method with soil moisture control was better than that of the baseline MOD16. In summary, the $f$ estimations proposed above performed well in their respective studies, but their applicability has not been sufficiently tested on barrens with sparse vegetation in arid or semi-arid basins, such as those found on the Tibetan Plateau.

Known as the "Asian Water Tower", the Tibetan Plateau (TP) is crucial to the development of the Asian monsoon and water and energy cycles (Yao et al., 2012). Although great efforts have been made to evaluate ET in the sub-basin of the TP over the past few years (Xue et al., 2013; Hu et al., 2018; Wang et al., 2018; Li et al., 2019), most studies have focused on the headwaters of rivers in eastern or southern TP and have ignored the central and western inland arid and semi-arid regions. Ma et al. (2020) provided some hourly land-atmosphere interaction observations

of inner regions with sparse vegetation; however, accurate soil evaporation estimates involving barrens remain a challenge. Li et al. (2014) reconstructed monthly ET estimates using the water balance method to evaluate five existing global ET products. They found that existing ET products were still not satisfactory for the Qaidam Basin and Qiangtang Plateau, two barrens-dominated sub-basins on the TP. In brief, the surface energy balance and land-atmosphere interaction mechanisms in alpine barren areas have not been explicitly revealed.

Based on two barrens-dominated sub-basins on the TP, the objectives of this study are: (1) to investigate the differences between the six existing soil evaporation algorithms and their applicability to alpine barren areas, (2) to improve the P-LSH algorithm by introducing two frameworks for quantifying moisture constraints to ET in terms of surface soil moisture and precipitation, respectively, and (3) to test the applicability of satellite soil moisture and precipitation data for improving ET retrieval and analyze the influence of soil moisture and precipitation

uncertainties on ET estimation on alpine barren areas.

## 2 Materials and study area

### 2.1 Study area

The Qaidam Basin is located in the northeastern TP (35°55'–39°10'N, 90°00'–98°20'E) and occupies an area of

257,768 km$^2$. The elevation of the Qaidam Basin is between 2,676 and 6,860 m, and the annual average temperature

ranges from -6.4 to 14.5 ℃. Saline lakes and deserts cover approximately one-quarter and one-third of the Qaidam

Basin, respectively. This region is thus very dry. The Qaidam Basin has a typical continental climate with an

average annual precipitation ranging from 29 to 387 mm, with approximately 80% of the precipitation occurring in

summer. Its drought conditions, high salinity, large diurnal and seasonal temperature ranges, and high ultraviolet

radiation make the basin unsuitable for living. According to the MODIS IGBP classification (Friedl et al., 2010),

79.1% of the Qaidam Basin is barren, 20.2% is grassland, and other land uses/land cover types represent less than

1%. Grassland is concentrated at the edge of the eastern and southern basins, whereas barren land is widely

distributed across the remaining basins (Fig. 1a).

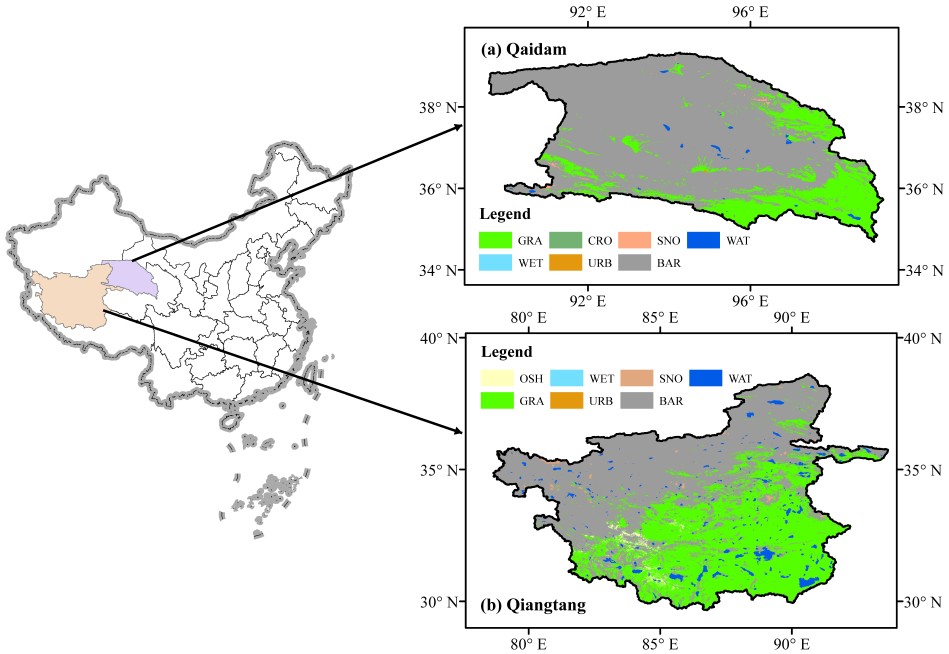

**Figure 1.** Locations and land cover/land use of **(a)** the Qaidam Basin and **(b)** the Qiangtang Plateau within China.
(OSH: open shrublands; GRA: grasslands; WET: wetlands; CRO: croplands; URB: urban and built-up lands; SNO:
snow and ice; BAR: barren; WAT: water bodies).

The Qiangtang Plateau is located in the central hinterland of the TP, close to the Qaidam Basin. It forms the main

feature of the TP with an area of 700,000 km$^2$. The average annual precipitation on the Qiangtang Plateau ranges

from 50 to 300 mm in solid forms, such as snow, graupel, and hail, with precipitation being concentrated in the

summer. The high altitude and inland surrounding high mountains make the Qiangtang Plateau a uniquely cold and



arid region with widely distributed permafrost. Similar to the Qaidam Basin, barrens account for the largest proportion of the Qiangtang Plateau, reaching 55.7%, whereas grassland and open water account for the second and third proportions, with values reaching 39.7% and 3.0%, respectively (Fig. 1b). The lakes on the Qiangtang Plateau

cover an area of 21,400 km$^2$, accounting for approximately a quarter of all lake areas in China. The unique geographical structure makes the Qiangtang Plateau an endorheic area, which is also true of the Qaidam Basin, where water is retained and no outflow to other external rivers or oceans occurs. In an endorheic basin, drainage converges into inner lakes or swamps and equilibrates through evaporation.

### 2.2 Satellite and meteorological inputs

Table 1 summarizes the datasets used in this study. All input datasets were bilinearly resampled from the original spatial resolution to a common 1/12° grid with a temporal resolution on a daily scale. The daily meteorological inputs required by remote sensing algorithms are derived from the China Meteorological Forcing Dataset (CMFD) (He et al., 2020), including air temperature (T), specific humidity (q), air pressure ($P_{air}$), wind speed ($u_m$), and precipitation (P). The dataset incorporates existing reanalysis datasets and in situ observations, and shows better

accuracy than existing reanalysis datasets (Yang et al., 2010; He et al., 2020). Radiation inputs come from the Clouds and the Earth's Radiant Energy System (CERES) SYN1deg radiative fluxes (Wielicki et al., 1996), which have provided continuous products since March 2000 with a resolution of 1° globally. In this study, we used all-sky incoming shortwave radiation and net radiation. The NDVI product used in this study is from the MODIS MOD13Q1 Version 6 (https://lpdaac.usgs.gov).

In our algorithm, the surface soil moisture and precipitation were used to restrain soil evaporation. We selected various surface soil moisture and precipitation datasets from satellites, microwave assimilation, machine-learning methods, and reanalysis. The surface soil moisture comes from five datasets including: (i) the soil moisture dataset of China based on microwave data assimilation (Yang et al., 2020) (denoted as $\theta_{Yang}$ in this study); (ii) the land surface soil moisture dataset of SMAP time-expanded daily 0.25° × 0.25° over the Qinghai-Tibet Plateau Area (Qu

et al., 2019) (denoted as $\theta_{Qu}$); (iii) the combined product from the European Space Agency's Climate Change Initiative (ESA CCI) Soil Moisture Version 06.1 (Gruber et al., 2019) (denoted as $\theta_{ESA\ CCI}$); (iv) Global Land Data Assimilation System (GLDAS) Noah Land Surface Version 2.1 (Rodell et al., 2004) (denoted as $\theta_{GLDAS\ Noah}$); and (v) the second Modern-Era Retrospective Analysis for Research and Applications (MERRA) Version 2 (Molod et al., 2015) (denoted as $\theta_{MERRA}$). The precipitation comes from five datasets including: (i) CMFD (denoted as $P_{CMFD}$); (ii)

Global Precipitation Measurement (GPM) IMERG Final Precipitation L3 Version 06 (Hou et al., 2014) (denoted as $P_{GPM}$); (iii) Multi-Source Weighted-Ensemble Precipitation (MSWEP) Version 2.8 (Beck et al., 2019) (denoted as $P_{MSWEP}$); (iv) GLDAS Noah (denoted as $P_{GLDAS\ Noah}$); and (v) MERRA (denoted as $P_{MERRA}$). All the above soil moisture and precipitation sequences were resampled to 1/12°.

Our algorithm adopts different parameterization schemes according to pixelated land cover, which comes from the

MODIS Land Cover Type Yearly L3 Global 500 m SIN Grid (MCD12Q1) (Friedl et al., 2010). The MCD12Q1 product provides land cover properties, which come from observations spanning one year from the Terra and Aqua satellites. Here, we used data from 2003 and regarded them as static values. We calculated the percentage of various





land covers for each pixel (1/12°), estimated the ET of various land covers, and then weighted each pixel by the percentage. Soil properties, including residual soil moisture and saturated water content, were obtained from the

China Dataset of Soil Hydraulic Parameters Pedotransfer Functions for Land Surface Modeling (Dai et al., 2013). We aggregated the dataset from the original 30" resolution to 1/12° using the arithmetic averaging method.

To evaluate the robustness and uncertainty of various remote-sensing algorithms, this study used reconstructed ET estimates derived from the terrestrial water balance method ($ET_{recon}$) as a benchmark. For endorheic basins, river discharge is zero, and ET is equal to the residue between precipitation and change in terrestrial water storage ($\Delta S$).

Based on this method, Li et al. (2014) established a monthly $ET_{recon}$ for the Qaidam Basin and Qiangtang Plateau from 2003 to 2012. The gridded precipitation data for this study were obtained from the National Meteorological Information Center of the China Meteorological Administration (CMA), and $\Delta S$ was obtained from Gravity Recovery and Climate Experiment (GRACE) land data.

**Table 1.** List of the forcing datasets used in this study with their original resolutions and references.

| Variable | Datasets | Temporal resolution | Spatial resolution | References |
|---|---|---|---|---|
| Air temperature Humidity Air pressure Wind speed | CMFD | 3 hours | 0.1° | (He et al., 2020) |
| Radiation | CERES SYN1deg | hourly | 1° | (Doelling et al., 2013) |
| NDVI | MOD13Q1 | 16-day | 250m | https://lpdaac.usgs.gov |
| Surface soil moisture | The Soil Moisture Dataset of China Based on Microwave Data Assimilation ($\theta_{Yang}$) | daily | 0.25° | (Yang et al., 2020) |
| | Land Surface Soil Moisture Dataset of SMAP Time-Expanded Daily 0.25°×0.25° over Qinghai-Tibet Plateau Area ($\theta_{Qu}$) | daily | 0.25° | (Qu et al., 2019) |
| | ESA CCI ($\theta_{ESA\ CCI}$) | daily | 0.25° | (Gruber et al., 2019) |
| | GLDAS Noah ($\theta_{GLDAS\ Noah}$) | 3 hours | 0.25° | (Rodell et al., 2004) |
| | MERRA ($\theta_{MERRA}$) | hourly | 0.5°×0.625° | (Molod et al., 2015) |
| Precipitation | CMFD ($P_{CMFD}$) | 3 hours | 0.1° | (He et al., 2020) |
| | GPM ($P_{GPM}$) | half-hourly | 0.1° | (Hou et al., 2014) |
| | MSWEP ($P_{MSWEP}$) | 3 hours | 0.1° | (Beck et al., 2019) |
| | GLDAS Noah ($P_{GLDAS\ Noah}$) | 3 hours | 0.25° | (Rodell et al., 2004) |
| | MERRA ($P_{MERRA}$) | hourly | 0.5°×0.625° | (Molod et al., 2015) |
| Land cover | MCD12Q1 | yearly | 500m | (Friedl et al., 2010) |
| Soil properties | A China Dataset of Soil Hydraulic Parameters Pedotransfer Functions for Land Surface Modeling | static | 30" | (Dai et al., 2013) |
| Reconstructed ET | - | monthly | Basin-scale | (Li et al., 2014) |


## 3 Methodology

### 3.1 Description of the Baseline Algorimth: P-LSH

The Process-based Land Surface Evapotranspiration/Heat Fluxes (P-LSH) algorithm (Zhang et al., 2010a; Zhang et al., 2015) is an ET algorithm evolved from the PM equation, in which canopy conductance comes from the Jarvis-Stewart formula (Jarvis, 1976; Stewart, 1988) and an empirical $g_0$-NDVI equation (Zhang et al., 2010a). The P-LSH algorithm distinguishes between open water and vegetation pixels using land cover classification. Vegetation pixels include canopy transpiration and soil evaporation, whereas open water pixels only contain water evaporation.

### (1) Canopy transpiration

The P-LSH algorithm calculates canopy transpiration ($E_c$: mm) by a modified PM equation:

$$\lambda E_c = \frac{\Delta A_c + \rho C_p VPD g_{a\_c}}{\Delta + \gamma (1 + g_{a\_c}/g_c)}, \tag{1}$$

where $\lambda$ (J kg$^{-1}$) is the latent heat of vaporization, $\Delta$ (Pa K$^{-1}$) is the slope of the curve relating saturated water vapor pressure to air temperature, VPD (Pa) is the vapor pressure deficit, $\rho$ (kg m$^{-3}$) is the air density, $C_p$ (J kg$^{-1}$ K$^{-1}$) is the specific heat capacity of air, $\gamma$ (-) is the psychrometric constant, $A_c$ (W m$^{-2}$) is the available energy component allocated to the canopy based on fractional vegetation cover, and $g_{a\_c}$ (m s$^{-1}$) is the aerodynamic conductance of the canopy. Based on various vegetation types, Zhang et al. (2010a) established an empirical relationship between the maximum canopy conductance ($g_0$: m s$^{-1}$) and NDVI based on observations from flux towers and reduced conductance from the maximum ($g_0$: m s$^{-1}$) to the actual value ($g_c$: m s$^{-1}$) through restraints from T (℃), VPD (Pa), and $CO_2$ (ppm). Feng et al. (2022) added incoming shortwave radiation and surface soil moisture to strengthen restraints on $g_c$ over three TP grasslands. More details regarding canopy transpiration are available in Feng et al. (2022) and Zhang et al. (2015).

### (2) Soil evaporation

The P-LSH algorithm combines the modified PM equation and complementary relationship hypothesis to quantify soil evaporation ($E_s$: mm) (Bouchet, 1963; Fisher et al., 2008), which can be expressed as:

$$E_s = f E_{eqs}, \tag{2}$$

$$\lambda E_{eqs} = \frac{\Delta A_s + \rho C_p VPD g_{a\_s}}{\Delta + \gamma g_{a\_s}/g_{totc}}, \tag{3}$$

$$f = RH^{\frac{VPD}{k}}, \tag{4}$$

where $f$ (-) is the moisture constraint, RH (-) is the relative humidity, k (Pa) is a parameter to fit the complementary relationship, $E_{eqs}$ (mm) is the equilibrium (i.e., potential) evaporation, $A_s$ (W m$^{-2}$) is the available energy component allocated to the soil surface, and $g_{totc}$ (m s$^{-1}$) is the corrected value of $g_{tot}$ (m s$^{-1}$) based on the standard temperature and pressure. In this study, the $g_{tot}$ term was expressed in the form of resistance $r_{tot}$ ($r_{tot} = 1/g_{tot}$: s m$^{-1}$) and $g_{a\_s}$ (m s$^{-1}$) is the aerodynamic conductance of the soil surface. More details regarding soil evaporation are available in Mu et al. (2007) and Zhang et al. (2010a).


**(3) Open water**

For open water pixels, the P-LSH algorithm uses the Penman equation rewritten by Shuttleworth (1993) to quantify
the effects of the surface wind speed on open water evaporation ($E_w$: mm). The surface resistance $r_s$ (s m$^{-1}$) is
assumed to be zero on the open water surface; therefore, the PM equation is revised as:

$$\lambda E_w = \frac{\Delta A + \rho C_p VPD g_{a\_w}}{\Delta + \gamma},$$ (5)

where A (W m$^{-2}$) is the available energy component for open water, following Zhang et al. (2010a). The $g_{a\_w}$ (m s$^{-1}$)
term is the aerodynamic conductance of the open water and is estimated by the wind speed:

$$g_{a\_w} = \frac{1 + 0.536 U_2}{4.72[ln(z_m/z_0)]^2},$$ (6)

where $U_2$ (m s$^{-1}$) is the 2 m height wind speed calculated from the reanalysis data and the vertical wind speed
function, $z_m$ (m) is the wind measurement height, and $z_0$ (m) is the aerodynamic roughness of the water surface,
which is set to 0.00137.

**3.2 Five existing soil evaporation algorithms**

In this study, we further selected the soil evaporation schemes from five existing ET algorithms, including the
Penman-Monteith-Leuning (PML) algorithm (Zhang et al., 2010b; Zhang et al., 2019), Global Land Evaporation
Amsterdam Model (GLEAM) algorithm (Martens et al., 2017), the Priestley Taylor-Jet Propulsion Laboratory (PT-
JPL) algorithm (Fisher et al., 2008), the Priestley Taylor-Yao (PT-Yao) algorithm (Yao et al., 2013), and the
Penman-Monteith-Brust (PM-Brust) algorithm (Brust et al., 2021).

**(1) PML soil evaporation algorithm**

The PML algorithm quantifies soil evaporation using the modified PT equation, which avoids the difficulty of
parameterizing the resistances in the PM equation (Zhang et al., 2010b; Zhang et al., 2019):

$$E_s = f E_{eqs,n},$$ (7)

$$\lambda E_{eqs,n} = \frac{\Delta A_s}{\Delta + \gamma},$$ (8)

where $A_s$ (W m$^{-2}$), $\Delta$ (Pa K$^{-1}$), and $\gamma$ (-) represent the same physical meanings as in Eq. (3). The moisture constraint $f$
(-) is estimated by the cumulative precipitation and equilibrium evaporation in the previous periods, without any
observation of soil moisture as input:

$$f = min\left(\frac{\sum_{n=1}^N P_n}{\sum_{n=1}^N E_{eqs,n}}, 1\right),$$ (9)

where $P_n$ (mm) and $E_{eqs,n}$ (mm) are the cumulative precipitation and equilibrium evaporation of the surface in the n[th]
period, respectively, and N is the number of periods.

**(2) GLEAM soil evaporation algorithm**

Similar to the PML algorithm, GLEAM takes the PT equation as the equilibrium soil evaporation and reduces it to
actual soil evaporation through the moisture constraint $f$ (Martens et al., 2017). The difference is that the GLEAM
algorithm estimates $f$ for tall canopies, short vegetation, and barren areas. For barren areas with sparse vegetation,



the surface soil moisture is linearized by the critical moisture level and residual soil moisture, and is then used to estimate soil evaporation, which is expressed as:

$$E_s = f E_p, \tag{10}$$

$$\lambda E_p = \alpha \frac{\Delta}{\Delta + \gamma} A_s, \tag{11}$$

$$f = 1 - \frac{\theta_c - \theta}{\theta_c - \theta_r}, \tag{12}$$

where $f$ (-) is the same as that in Eq. (9) to explain the restraints of the suboptimal environment on soil evaporation; $E_p$ (mm) is the potential soil evaporation; $\alpha$ (-) is the PT dimensionless coefficient, and 1.26 for barrens; $\theta$ ($cm^3$ $cm^{-3}$) is the actual surface soil moisture; $\theta_c$ ($cm^3$ $cm^{-3}$) is the critical moisture level and is set as $\theta_c = 0.75\theta_s$ following Zhu et al. (2013), where $\theta_s$ ($cm^3$ $cm^{-3}$) is the saturated water content and $\theta_r$ ($cm^3$ $cm^{-3}$) is the residual soil moisture.

**(3) PT-JPLsoil evaporation algorithm**

The PT-JPL algorithm uses the same equilibrium soil evaporation as the GLEAM algorithm, with the difference being in the $f$ estimation (Fisher et al., 2008). In the PT-JPL algorithm, $f$ is constituted by $f_{SM}$ (-) and $f_{wet}$ (-), where $f_{SM}$ comes from RH and VPD (the same as in the P-LSH algorithm), whereas $f_{wet}$ is only determined by RH:

$$f_{SM} = RH^{\frac{VPD}{k}}, \tag{13}$$

$$f_{wet} = RH^4, \tag{14}$$

$$E_s = [f_{wet} + f_{SM}(1 - f_{wet})]E_p, \tag{15}$$

where $k$ (Pa) has the same meaning as that in Eq. (4), and $E_p$ (mm) is the equilibrium soil evaporation calculated using Eq. (11).

**(4) PT-Yao soil evaporation algorithm**

Yao et al. (2013) used the diurnal land surface temperature range (DTsR: ℃) and air temperature range (DTaR:℃) to simplify the calculation of the Apparent Thermal Inertia (ATI: $℃^{-1}$) for $f_{SM}$ (-) estimation with equilibrium soil evaporation using the PT equation, same as the GLEAM and PT-JPL algorithms:

$$f_{SM} = ATI^k = \left(\frac{1}{DT}\right)^{DT/DT_{max}}, \tag{16}$$

where $DT_{max}$ (℃) is defined as the maximum daily temperature range (DT:℃), which reflects the relative sensitivity to changes in the daily temperature range and is set as a constant ($DTaR_{max} = 40℃$, $DTsR_{max} = 60℃$). Yao et al. (2013) showed that the performances of soil evaporation from DTaR and DTsR are similar; therefore, in this study, we only used DTaR for $f_{SM}$ estimation.

**(5) PM-Brust soil evaporation algorithm**

The PM-Brust algorithm (Brust et al., 2021) originated from the MOD16 algorithm that is based on the PM equation (Mu et al., 2011). The equilibrium soil evaporation in the PM-Brust algorithm is similar to Eq. (3), with the resistance estimations slightly different from those of the P-LSH algorithm. The PM-Brust algorithm assumes that the boundary layer resistance is equal to the total aerodynamic resistance ($r_{tot}$: s $m^{-1}$), which is determined by VPD





and four biome-specific constants, including maximum resistance ($rbl_{max}$: s m$^{-1}$), minimum resistance ($rbl_{min}$: s m$^{-1}$), VPD at which canopy stomata are completely open ($VPD_{open}$: Pa), and VPD at which canopy stomata are completely close ($VPD_{close}$: Pa). In contrast, the P-LSH algorithm assumes that the boundary layer resistance and total aerodynamic resistance are biome-specific constants. Brust et al. (2021) estimated $f_{SM}$ with a more direct soil moisture control outline (i.e., REW):

$$f_{SM} = REW = \frac{\theta - \theta_{min}}{\theta_{max} - \theta_{min}}, \tag{17}$$

$$\lambda E_{eqs} = \frac{\Delta A_s + \rho C_p (1 - f_c) VPD g_{a\_s}}{\Delta + \gamma \times g_{a\_s}/g_{totc}}, \tag{18}$$

$$E_s = [f_{wet} + f_{SM}(1 - f_{wet})]E_{eqs}, \tag{19}$$

where REW (-) is the relative extractable water, $\theta$ (cm$^3$ cm$^{-3}$) is the surface soil moisture, $\theta_{min}$ (cm$^3$ cm$^{-3}$) and $\theta_{max}$ (cm$^3$ cm$^{-3}$) are the minimum and maximum values of $\theta$ for the period of record, respectively, and $f_c$(-) is the vegetation cover fraction.

### 3.3 Improvements of the P-LSH soil evaporation algorithm

We attempted two strategies to improve soil evaporation in the P-LSH algorithm. One strategy was to directly control $f$ through the surface soil moisture as follows:

$$\lambda E_s = \frac{\theta - \theta_{min}}{\theta_{max} - \theta_{min}} \frac{\Delta A_s + \rho C_p VPD g_{a\_s}}{\Delta + \gamma g_{a\_s}/g_{totc}}, \tag{20}$$

where each item has the same meaning as that in Eq.(3) and Eq.(17).

The second strategy was to use the ratio of cumulative precipitation to equilibrium evaporation in the previous periods to quantify moisture constraint, with equilibrium evaporation estimated by the modified PM equation as follows:

$$\lambda E_s = min\left(\frac{\sum_{n=1}^{N} P_n}{\sum_{n=1}^{N} \frac{\Delta A_s + \rho C_p VPD g_{a\_s}}{\Delta + \gamma g_{a\_s}/g_{totc}}}, 1\right) \frac{\Delta A_s + \rho C_p VPD g_{a\_s}}{\Delta + \gamma g_{a\_s}/g_{totc}}, \tag{21}$$

where each item has the same meaning as that in Eq.(3) and Eq.(9).

We combined the vegetation evapotranspiration and open water evaporation components with the new soil evaporation component based on the first strategy to form an improved P-LSH algorithm, which is called P-LSH$_\theta$. Similarly, we built the second improved P-LSH algorithm based on the second strategy (hereafter it is called P-LSH$_P$). By contrast, the original P-LSH soil evaporation algorithm is called P-LSH$_{ori}$ in this study.

### 3.4 Evaluation of algorithm performance

Because we don't have direct observation of soil evaporation, we have to rely on the $ET_{recon}$ as the benchmark to assess our improved soil evaporation algorithms and their associated ET retrieval algorithms. Therefore, we need to assemble the soil evaporation algorithm with the vegetation evapotranspiration and water evaporation algorithms to form a complete ET retrieval algorithm to estimate ET. To this end, we coupled the vegetation evapotranspiration scheme and water evaporation scheme of the P-LSH algorithm with the six existing soil evaporation algorithms (namely, the soil evaporation algorithms of the PML, GLEAM, PT-JPL, PT-Yao, PM-Brust, and P-LSH$_{ori}$) to produce six ET retrieval algorithms (i.e., A1 to A6 of Table 2 and Fig. 2). Therefore, A1, A2, A3, A4, A5, and A6





are comparable to P-LSH$_\theta$, and P-LSH$_P$ because the only difference between these algorithms is their soil evaporation component. We then compared the performances of A1, A2, A3, A4, A5, A6, P-LSH$_\theta$, and P-LSH$_P$ for barren areas from January 2003 to August 2011 using $ET_{recon}$ as the benchmark.

**Table 2.** Combinations of the six existing soil evaporation algorithms with the P-LSH vegetation evapotranspiration
and water evaporation schemes.

| Vegetation evapotranspiration and water evaporation algorithm | Barren evaporation algorithm | Coupling algorithm | ET estimation |
|---|---|---|---|
| | PML | A1 | $ET_{A1}$ |
| | GLEAM | A2 | $ET_{A2}$ |
| P-LSH | PT-JPL | A3 | $ET_{A3}$ |
| | PT-Yao | A4 | $ET_{A4}$ |
| | PM-Brust | A5 | $ET_{A5}$ |
| | P-LSH | A6 | $ET_{A6}$ |

The total ET in a pixel is expressed as:

$$E = \sum_i E_i a_i, \tag{22}$$

where i represents the i[th] land cover in the basin. We ignored land cover that accounted for less than 1%, so there
were grasslands, barrens, and open water for the Qaidam Basin and open shrublands, grasslands, barrens, and open water for the Qiangtang Plateau. The $E_i$ (mm) is the evapotranspiration estimated from the i[th] land cover, and $a_i$ (-) is the proportion of the i[th] land cover in a pixel. The open shrubland and open water pixels take the vegetation evapotranspiration scheme and water evaporation scheme from the P-LSH algorithm following Zhang et al. (2010a) and Zhang et al. (2015), and the grassland pixels take the vegetation evapotranspiration scheme from the revised P-
LSH algorithm following Feng et al. (2022). For barrens, we took the assumption that there was no canopy transpiration, and the performance of the six existing and two improved soil evaporation algorithms were compared. A flowchart of the total ET estimation for the basin is shown in Fig. 2.





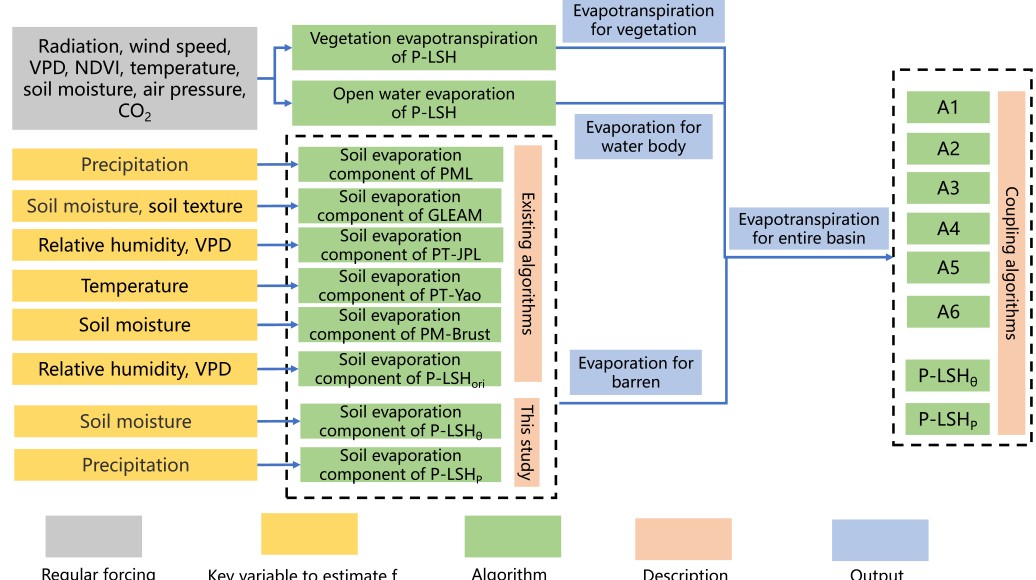

**Figure 2.** Flowchart of the gridded evapotranspiration estimation for a basin used in this study.


We chose the root mean square error (RMSE), coefficient of determination ($R^2$), deviation (bias), and Nash-Sutcliffe efficiency coefficient (NSE) to quantify the performances of remote sensing algorithms compared with the $ET_{recon}$:

$$RMSE = \sqrt{\frac{1}{T}\sum_{i=1}^{T}(O_i - S_i)^2},$$ (23)

$$R^2 = \left(\frac{\sum_{i=1}^{T}(O_i - \bar{O})(S_i - \bar{S})}{\sqrt{\sum_{i=1}^{T}(O_i - \bar{O})^2}\sqrt{\sum_{i=1}^{T}(S_i - \bar{S})^2}}\right)^2,$$ (24)

$$Bias = \frac{1}{T}\sum_{i=1}^{T}(O_i - S_i),$$ (25)

$$NSE = 1 - \frac{\sum_{i=1}^{T}(O_i - S_i)^2}{\sum_{i=1}^{T}(O_i - \bar{O})^2},$$ (26)

where T is the number of months in the period of record, O is the reconstructed ET, S is the simulated ET, $\bar{O}$ is the average of all reconstructed values $O_i$, and $\bar{S}$ is the average of all simulated $S_i$.

## 4 Results

### 4.1 Performance of existing soil evaporation algorithms

We estimated the daily and 1/12° pixel ET in the Qaidam Basin and Qiangtang Plateau from January 2003 to August 2011 using the six coupling algorithms listed in Table 2. All daily and gridded estimates were aggregated to monthly and basin scales to match $ET_{recon}$. Generally, the ET estimates derived from the six coupling algorithms showed large differences. In the Qaidam Basin, the ET estimates of the A1 algorithm ($ET_{A1}$) and the A5 algorithm ($ET_{A5}$)

demonstrated good consistency with the $ET_{recon}$, while the ET estimates of the A3 algorithm ($ET_{A3}$) and the A6



algorithm ($ET_{A6}$) matched the worst. The $ET_{A1}$ estimates performed best among all the existing algorithms (Fig. 3a), with an RMSE of 4.06 mm month$^{-1}$, an NSE of 0.88, and an R$^2$ of 0.92. The ET estimates of the A2 algorithm ($ET_{A2}$) with a linear formula for $f$ were well-simulated for low intervals and were always underestimated for the middle and high intervals (Figs. 3b and 4a). Parameter k in the PT-JPL algorithm was a biome-specific constant and took the same value for all barren pixels, set to 926 Pa, which was calibrated by the $ET_{recon}$. Although the parameter k has been calibrated, $ET_{A3}$ still could not accurately describe the seasonal variability of ET (Figs. 3c and 4a), mainly because of errors involving $f$ estimates derived by RH and VPD. The medium ET estimates of the A4 algorithm ($ET_{A4}$) were always overestimated for the Qaidam Basin (Fig. 3d), which specifically occurred in spring (Fig. 4a). In the PM-Brust method, four biome-specific constants ($rbl_{max}$, $rbl_{min}$, $VPD_{close}$, and $VPD_{open}$) for the $r_{tot}$ estimation were set to 500 s m$^{-1}$, 200 s m$^{-1}$, 4200 Pa, and 650 Pa, respectively, for the Qaidam Basin. The $ET_{A5}$ presented good performance (Fig. 3e), with an RMSE of 4.36 mm month$^{-1}$, an NSE of 0.87, and an R$^2$ of 0.88. The $ET_{A6}$ estimates used RH and VPD to estimate $f$, with parameter k of 359.1 Pa and $r_{tot}$ of 462.4 s m$^{-1}$ following Feng et al. (2022). However, $ET_{A6}$ could not adequately describe seasonal variability (Figs. 3f and 4a) in the Qaidam Basin, and seasonal mean values also varied by a large margin compared with $ET_{recon}$ (Fig. 4a).




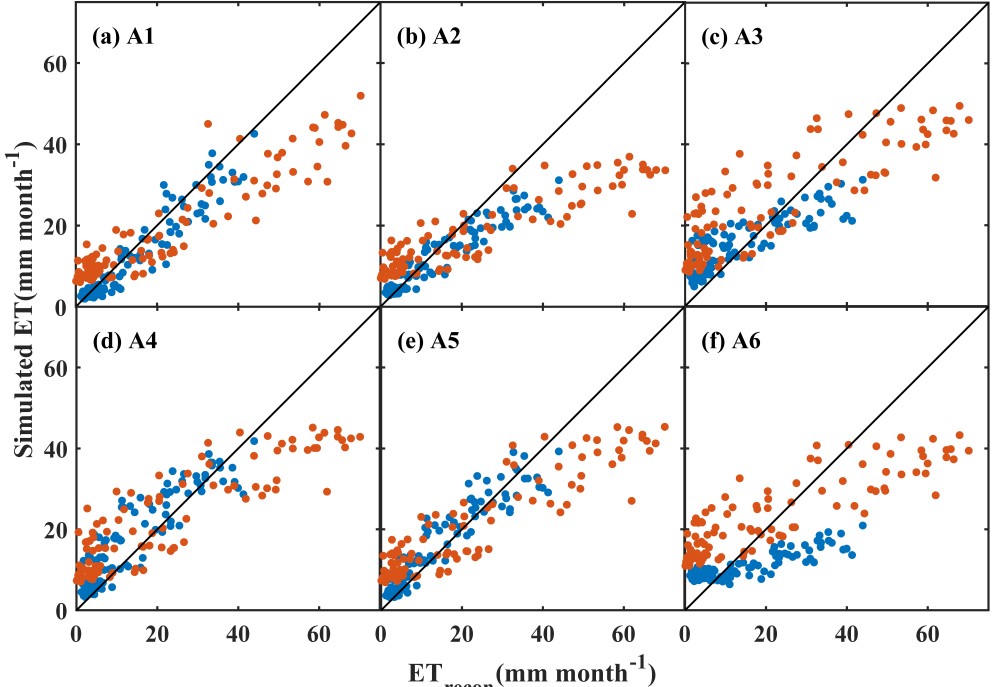

**Figure 3.** Simulated ET derived from the six existing coupling algorithms versus the $ET_{recon}$. The blue dots represent results for Qaidam Basin and the red for Qiangtang Plateau.

On the Qiangtang Plateau, almost all algorithms overestimated ET for barren areas in spring and winter and



underestimated ET in summer and autumn (Fig. 4b). The multi-year average $ET_{recon}$ in spring and winter was 6.3 mm month$^{-1}$, while the multi-year average ET derived from six coupling remote sensing algorithms was 14.4 ± 6.8 mm month$^{-1}$. The multi-year average $ET_{recon}$ in summer and autumn was 38.3 mm month$^{-1}$, and it was 28.2 ± 12.1 mm month$^{-1}$ from six remote sensing algorithms. In the comparison of the six algorithms, the $ET_{A1}$ estimates

still performed best among all algorithms, with an RMSE of 11.14 mm month$^{-1}$, and $ET_{A2}$ estimates performed the worst, with an RMSE of 14.46 mm month$^{-1}$. The biome-specific constant, $k$, in the PT-JPL algorithm was recalibrated to 566 Pa using $ET_{recon}$ for the Qiangtang Plateau. In the PT-JPL and P-LSH$_{ori}$ algorithms, unreasonable $f$ estimates also led to the homogenization of strong seasonal variability (Figs. 3c, 3f, and 4b). Similar to the Qaidam Basin, the $ET_{A4}$ estimates showed moderate performance for the Qiangtang Plateau (Fig. 3d), and the $ET_{A5}$ estimates

showed good performance next to $ET_{A1}$, with an RMSE of 11.42 mm month$^{-1}$, an NSE of 0.72, and an R$^2$ of 0.85.

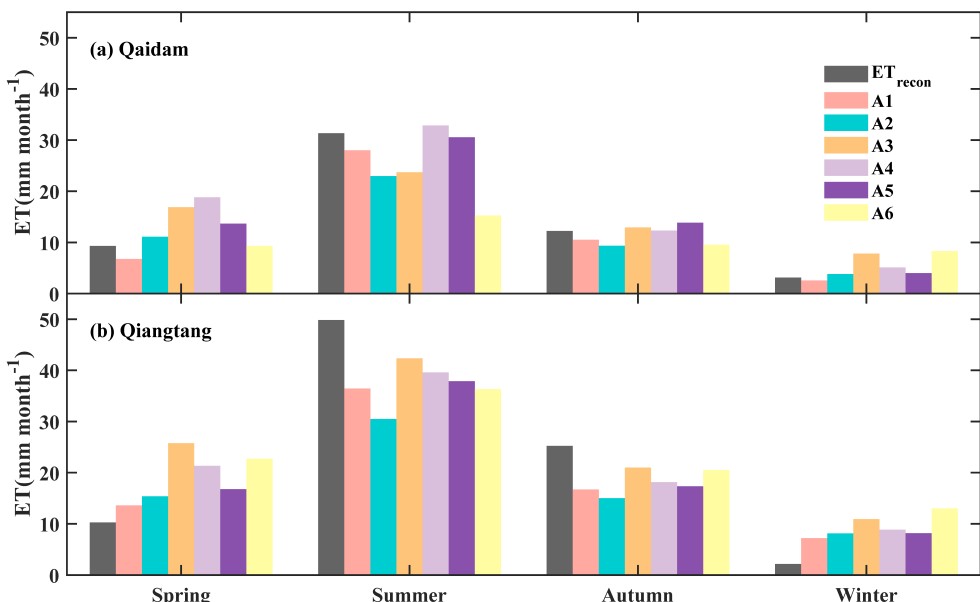

**Figure 4.** Seasonal average $ET_{recon}$ and ET estimates derived from six existing coupling algorithms for **(a)** the Qaidam Basin and **(b)** the Qiangtang Plateau.

We calculated the moisture constraint $f$ in the soil evaporation of each pixel and used the same method as ET to aggregate $f$ at the monthly and basin scales. The $f$ estimates derived from various algorithms are shown in Fig. 5. The $f$ estimates of the PML algorithm ($f_{PML}$) were high in summer and low in winter, with distinct seasonality in both basins, with small peaks occasionally appearing in winter. The $f$ estimates of the GLEAM algorithm ($f_{GLEAM}$) hardly showed seasonality and were always low in both basins, which was the main reason for the poor performance of

$ET_{A2}$. Compared with $f_{PML}$ and $f_{GLEAM}$ estimates, the $f$ estimates of the PT-Yao algorithm ($f_{PT-Yao}$) were overestimated in spring and winter, partly causing the overestimation of $ET_{A4}$, and this overestimation was larger than that of $f_{PML}$ (Figs. 3d and 4). Considering the positive relationship between precipitation and soil moisture, the $f$ estimates of the



PM-Brust algorithm ($f_{PM-Brust}$) from soil moisture and the $f_{PML}$ estimates from precipitation showed great consistency, with correlation coefficients of 0.86 and 0.85 (p < 0.001) for the Qaidam Basin and Qiangtang Plateau, respectively.

However, the $f_{PM-Brust}$ estimates were higher overall than $f_{PML}$ in spring and winter and hardly ever close to zero, indicating that the soil moisture sequences over basins seldom reached their minimum at the same time. In addition, compared with $f_{PML}$, the overestimation of $f_{PM-Brust}$ was also a reason for the overestimation of $ET_{A5}$ in spring and winter (Figs. 4a and 5). The PT-JPL and P-LSH$_{ori}$ algorithms shared a similar $f$ estimation and had the same temporal characteristics, with high values in winter and low values in summer, which showed the opposite seasonal

variability to soil moisture (expressed in the form of $f_{PM-Brust}$). Therefore, the performances of $ET_{A3}$ and $ET_{A6}$ were unsatisfactory. This is because the VPD sequence for both basins on the TP had stronger seasonality (high in summer and low in winter) compared to the milder RH. Although ET estimates derived from the PT-JPL and P-LSH$_{ori}$ algorithms have been well-validated in some flux towers (Fisher et al., 2008; Zhang et al., 2010a; Mu et al., 2011), this method is no longer applicable because of the unique meteorology of the TP (mainly manifested in the

seasonality of RH and VPD) and the possibly decoupling of VPD and soil moisture on a daily scale.

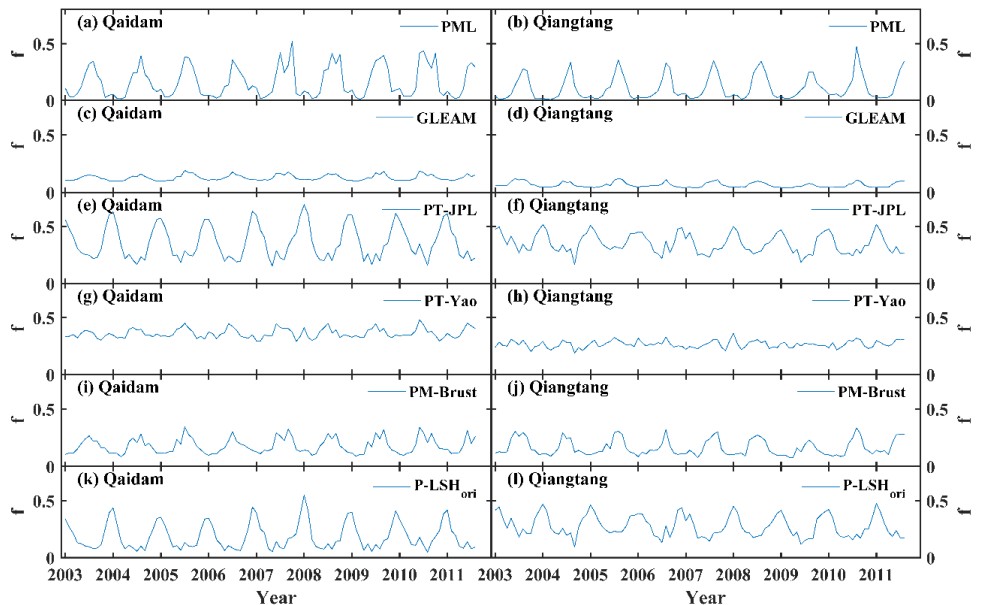

**Figure 5.** The monthly average $f$ derived from the six existing soil evaporation algorithms for the Qaidam Basin and the Qiangtang Plateau

## 4.2 Performance of the two improved P-LSH algorithms

Because of the good performance of surface soil moisture and precipitation in moisture constraints of the land surface, both were used to improve the P-LSH algorithm, called P-LSH$_θ$ and P-LSH$_P$. The soil moisture sequence was obtained from the assimilation-based $θ_{Yang}$, and the precipitation sequence was obtained from the satellite-based $P_{GPM}$. In the original P-LSH algorithm, $r_{tot}$ was a biome-specific constant sensitive to soil evaporation (Feng et al.,





2022). Therefore, we separately calibrated $r_{tot}$ for both basins in the P-LSH$_\theta$ algorithm, and the calibration was

completed using the loop of the parameter, with the RMSE as the objective function. The calibrated $r_{tot}$ values were 575 and 290 sm$^{-1}$ for the Qaidam Basin and the Qiangtang Plateau, respectively. The ET estimates derived from P-LSH$_\theta$ ($ET_{P\text{-}LSH\_\theta}$) matched well with the $ET_{recon}$ and captured the strong seasonality of both basins (Fig. 6). The P-LSH$_\theta$ algorithm had advantages in normalized standard deviation and centered RMSE, with values of 0.80 and 0.40, while they were $0.61 \pm 0.08$ and $0.55 \pm 0.08$ of existing coupling algorithms in Sect. 4.1 (Fig. 7). The $r_{tot}$ value in the

P-LSH$_P$ algorithm for each basin was set the same as that in the P-LSH$_\theta$ algorithm. The ET estimates derived from the P-LSH$_P$ ($ET_{P\text{-}LSH\_P}$) were similar to $ET_{P\text{-}LSH\_\theta}$ and showed a better simulation of the Qaidam Basin, especially the simulations of low values in spring and winter (Fig. 6). However, the $ET_{P\text{-}LSH\_P}$ estimates were always underestimated on the Qiangtang Plateau, much lower than the $ET_{recon}$ in summer, which may have been caused by the error of the GPM satellite precipitation on the Qiangtang Plateau (Li et al., 2020) (also see Sect. 4.3).

Nevertheless, $ET_{P\text{-}LSH\_P}$ estimates still performed well, second only to $ET_{P\text{-}LSH\_\theta}$ among all algorithms for both basins (Fig. 7). In summary, the P-LSH$_\theta$ and P-LSH$_P$ algorithms for both basins showed better performance than the existing algorithms in Sect. 4.1.

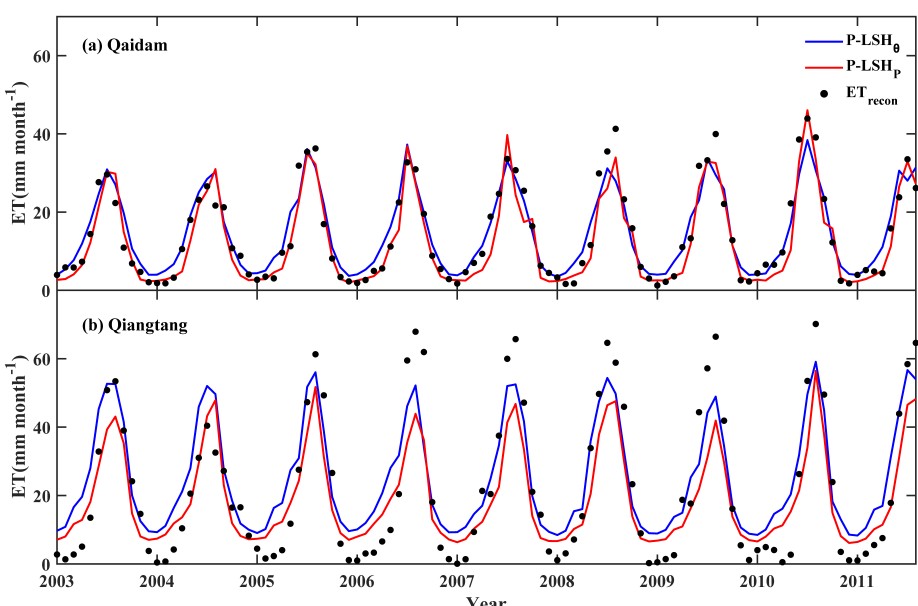

**Figure 6.** Comparisons of the monthly regional average ET estimates derived from two improved retrieval
algorithms (P-LSH$_\theta$ and P-LSH$_P$) with the $ET_{recon}$ for **(a)** the Qaidam Basin and **(b)** the Qiangtang Plateau.

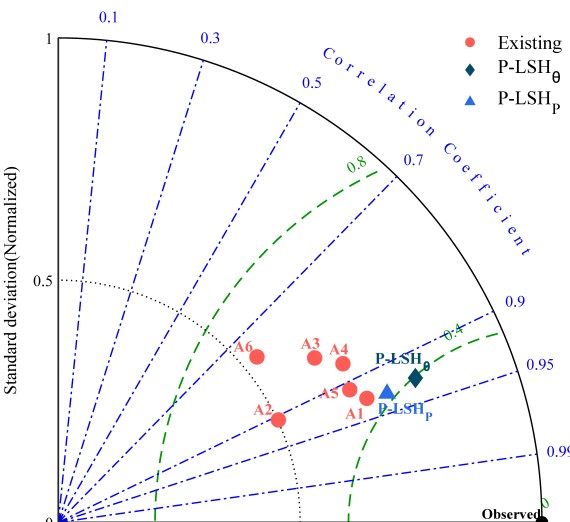

**Figure 7.** Taylor diagram comparing the retrieved ET by the six existing coupling algorithms and the two improved retrieval algorithms (P-LSH$_\theta$ and P-LSH$_P$) in the two basins. The green dashed line represents the centered root mean square error.


The multiyear average annual $ET_{P-LSH\_\theta}$ and $ET_{P-LSH\_P}$ estimates for both basins are shown in Fig. 8. The estimations of the two algorithms shared a similar spatial pattern, with a decreasing trend from the southeastern to northwestern basins. From the perspective of the regional average, $ET_{P-LSH\_\theta}$ and $ET_{P-LSH\_P}$ were 177 and 148 mm for the Qaidam Basin, respectively, and 300 and 232 mm for the Qiangtang Plateau, respectively. However, in the central Qaidam

Basin and northwest of the Qiangtang Plateau, the $ET_{P-LSH\_P}$ estimates were generally lower than those of $ET_{P-LSH\_\theta}$, and these underestimations existed in almost all seasons (Fig. 9). This underestimation was little in winter because both precipitation and soil moisture in winter were low, and the spatial differences between $ET_{P-LSH\_P}$ and $ET_{P-LSH\_\theta}$ almost disappeared.



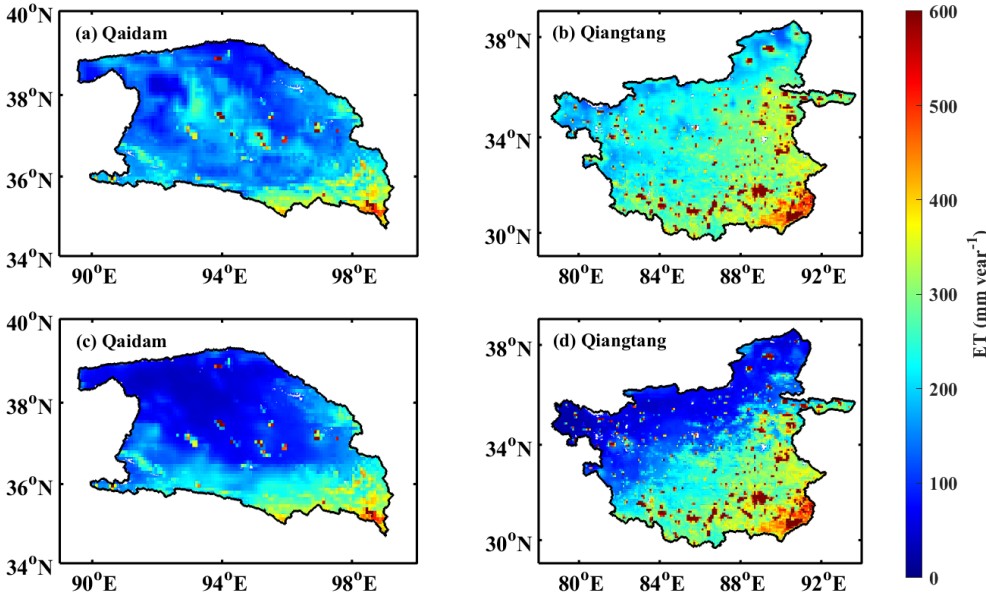

**Figure 8.** The spatial distributions of multi-year (2003.1–2011.8) average annual ET derived from **(a, b)** the P-LSH$_\theta$
and **(c, d)** P-LSH$_P$ for **(a, c)** the Qaidam Basin and **(b, d)** the Qiangtang Plateau.

Fig. 9 shows the multi-year spring (March, April, and May), summer (June, July, and August), autumn (September,
October, and November), and winter (December, January, and February) $ET_{P\text{-}LSH\_\theta}$ and $ET_{P\text{-}LSH\_P}$ in both basins. The
pattern of seasonal estimates was similar to that of the annual values. Generally, the ET in autumn was higher than
that in spring, with 71% of $ET_{P\text{-}LSH\_\theta}$ and 97% of $ET_{P\text{-}LSH\_P}$ for the Qaidam Basin and 72% of $ET_{P\text{-}LSH\_\theta}$ and 85% of
$ET_{P\text{-}LSH\_P}$ for the Qiangtang Plateau (percentage represents the number of pixels accounting for the basin). The
multi-year seasonal $ET_{P\text{-}LSH\_\theta}$ and $ET_{P\text{-}LSH\_P}$ averaged over the Qaidam Basin were 36, 88, 40, and 13 mm, and 20, 87,
33, and 8 mm for spring, summer, autumn, and winter, respectively. The multi-year seasonal $ET_{P\text{-}LSH\_\theta}$ and $ET_{P\text{-}LSH\_P}$
averaged over the Qiangtang Plateau were 61, 142, 68, and 29 mm, and 41, 114, 55, and 22 mm for spring, summer,
autumn, and winter, respectively.



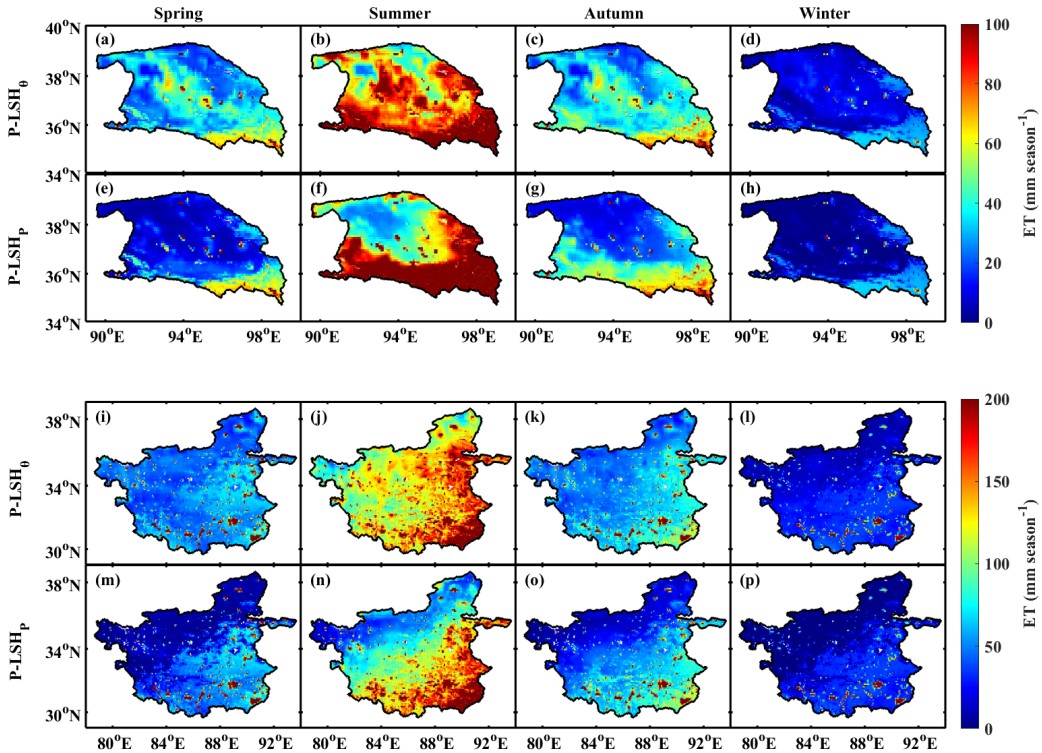

**Figure 9.** The spatial distributions of multi-year (2003.1-2011.8) seasonal ET derived from **(a-d, i-l)** P-LSH$_\theta$ and **(e-h, m-p)** P-LSH$_P$ for **(a-h)** the Qaidam Basin and **(i-p)** the Qiangtang Plateau.


### 4.3 Uncertainty of soil moisture and precipitation influence on soil evaporation

Surface soil moisture and precipitation were used to improve the P-LSH algorithm (Sect. 4.2). The two improved algorithms were highly dependent on high-quality gridded data; therefore, we selected five surface soil moisture and five precipitation datasets to discuss the impact of moisture constraint uncertainty on soil evaporation in the P-LSH$_\theta$

and P-LSH$_P$ algorithms. The daily and 1/12° pixel soil evaporation estimates for both basins were estimated and aggregated to monthly and basin scales. We calculated the coefficient of variation (*Cv*) between five barren evaporation estimates from the P-LSH$_\theta$ and five barren evaporation estimates from the P-LSH$_P$ algorithms, where the non-barren estimate was masked (hereafter $E_{s\_P\text{-}LSH\_\theta}$ and $E_{s\_P\text{-}LSH\_P}$, where the subscript s denotes soil evaporation for barrens). The *Cv* is defined as the ratio of the standard deviation to the mean. In the following part,

we further discussed the impacts of temporal and spatial uncertainties in soil moisture and precipitation on soil evaporation estimates.

To quantify the temporal uncertainties in soil moisture and precipitation and their resultant ET uncertainties, we calculated the multi-monthly (i.e., monthly average from 2003 to 2011) values for every soil moisture and



precipitation dataset and its associated ET estimate on a grid-cell basis. There was a clear spatial pattern of the $Cv$

among multi-monthly (i.e., monthly average from 2003 to 2011) average $E_{s\_P\text{-}LSH\_\theta}$ and $E_{s\_P\text{-}LSH\_P}$ from various

datasets. The $Cv$ of $E_{s\_P\text{-}LSH\_\theta}$ showed little variation in both basins (Fig. 10a and 10b). The $Cv$ of $E_{s\_P\text{-}LSH\_\theta}$ on the

Qaidam Basin ranged from 0.05 to 0.65 with a mean of 0.29, and ranged from 0.03 to 0.71 with a mean value of

0.29 on the Qiangtang Plateau. In contrast, the $Cv$ of $E_{s\_P\text{-}LSH\_P}$ was not consistent in both basins. In the Qaidam

Basin, the $Cv$ of $E_{s\_P\text{-}LSH\_P}$ was at a lower level (Fig. 10c), with an average of 0.29, which was comparable to that of

$E_{s\_P\text{-}LSH\_\theta}$. On the Qiangtang Plateau, the $Cv$ of $E_{s\_P\text{-}LSH\_P}$ increased from the southeast to the northwest of the basin

(Fig. 10d), with an average of 0.46, which was higher than that of $E_{s\_P\text{-}LSH\_\theta}$.

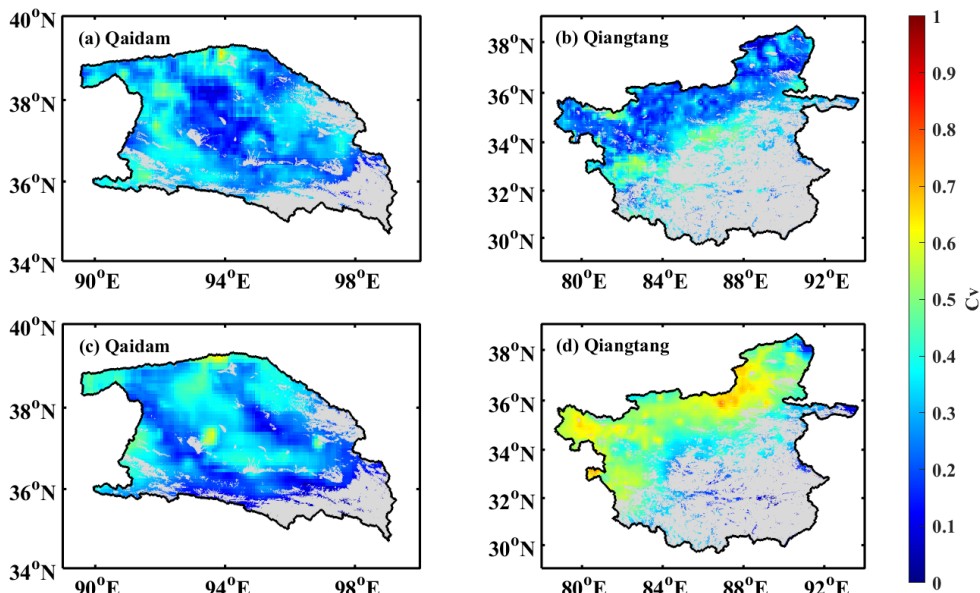

**Figure 10.** Maps of the $Cv$s of monthly average soil evaporation from 2003 to 2011 for barrens estimated by (**a, b**) the P-LSH$_\theta$ algorithm driven by five soil moisture datasets and (**c, d**) the P-LSH$_P$ algorithm driven by five
precipitation datasets in (**a, c**) the Qaidam Basin and (**b, d**) the Qiangtang Plateau. The gray indicates the non-barren areas within the basin.

To further distinguish the impact of the datasets and algorithm structure on barren evaporation estimates, we

compared the variation for various surface soil moisture and precipitation datasets, as shown in Fig. 11. The surface

soil moisture had high uncertainty in the central and northern Qaidam Basin and western Qiangtang Plateau (Fig.

11a and 11b), but these uncertainties were not reflected in $E_{s\_P\text{-}LSH\_\theta}$, indicating that the moisture constraint

calculated by Eq. (20) reduced the uncertainty of soil moisture and, instead, focused more on the relative changes in

the soil moisture of each dataset. The $Cv$ of the precipitation showed a similar spatial distribution to that of $E_{s\_P\text{-}LSH\_P}$

in both basins, and their correlation coefficients were 0.93 and 0.98 ($p < 0.001$) for the Qaidam Basin and Qiangtang

Plateau, respectively, indicating that the characteristics of precipitation were almost completely transferred to the

$E_{s\_P\text{-}LSH\_P}$ through Eq. (21). In contrast, the correlation coefficients of the $Cv$ of soil moisture and $E_{s\_P\text{-}LSH\_\theta}$ were only

0.33 and 0.46 ($p < 0.001$) for the two basins.



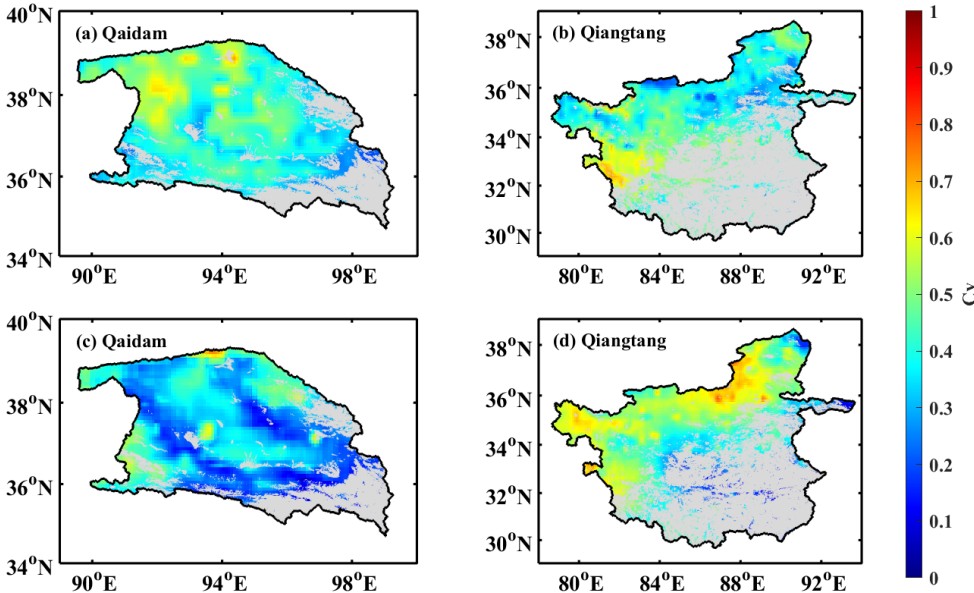

**Figure 11.** Maps of the *Cv*s of **(a, b)** five soil moisture datasets and **(c, d)** five precipitation datasets in **(a, c)** the
Qaidam Basin and **(b, d)** the Qiangtang Plateau. Each soil moisture or precipitation dataset is the monthly average
value from 2003 to 2011. The gray indicates the non-barren areas within the basin.

From the perspective of the regional average, we used *Cv* to express uncertainty, considering the magnitude between

soil moisture and precipitation. There were some uncertainties in various soil moisture datasets, especially in spring,

autumn, and winter (Fig. 12a and 12b), with *Cv* of 0.41 ± 0.07 and 0.41 ± 0.08 for the Qaidam Basin and Qiangtang

Plateau, respectively. The uncertainty of the peaks in summer tended to be much lower and always at a lower level

soil moisture values. The uncertainty among various precipitation datasets was comparable with soil moisture, with

*Cv* of 0.36 ± 0.20 and 0.55 ± 0.22 for the Qaidam Basin and Qiangtang Plateau. The *Cv* of the precipitation had a

similar temporal pattern, similar as soil moisture, low in summer, and high in other seasons (Fig. 12c and 12d). In

terms of $E_{s\_P\text{-}LSH\_\theta}$ and $E_{s\_P\text{-}LSH\_P}$, and considering the same object, we used the interval length of various estimates to

express uncertainty. Overall, the uncertainty of $E_{s\_P\text{-}LSH\_P}$ was lower than that of $E_{s\_P\text{-}LSH\_\theta}$, especially in spring and

winter in both basins (Fig. 12e and 12f). The interval length of $E_{s\_P\text{-}LSH\_P}$ were 4.94 ± 3.63 and 14.61 ± 10.45 mm

month$^{-1}$, and were 11.41 ± 5.91 and 16.92 ± 7.01 mm month$^{-1}$ in $E_{s\_P\text{-}LSH\_\theta}$ for the Qaidam Basin and Qiangtang

Plateau, respectively. On the Qiangtang Plateau, the higher uncertainty of the precipitation datasets led to a larger

interval length in the estimation of $E_{s\_P\text{-}LSH\_P}$ compared with the Qaidam Basin, yet this uncertainty was still smaller

than that of $E_{s\_P\text{-}LSH\_\theta}$.



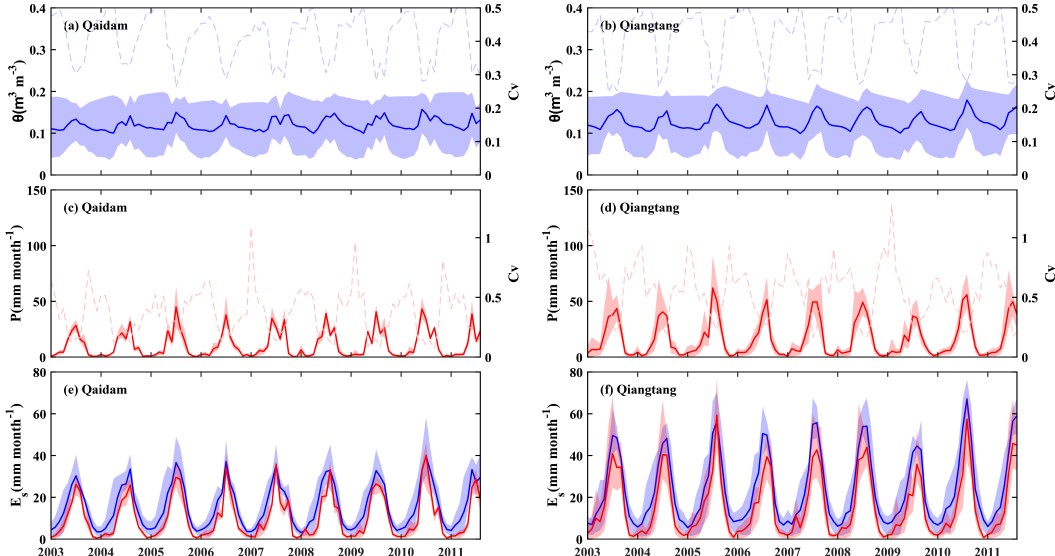

**Figure 12.** Monthly regional average **(a-b)** soil moisture datasets, **(c-d)** precipitation datasets, **(e-f)** soil evaporation estimates and their $Cv$s over barrens of the Qaidam Basin and the Qiangtang Plateau. The shades indicate the range determined by five datasets/estimates and solid lines represent the mean of them, depending on the left y-axis. The light dashed lines represent $Cv$s of five datasets/estimates, depending on the right y-axis. Blue represents results of soil moisture datasets or $E_s$ estimates derived from soil moisture, and red represents results of precipitation datasets or $E_s$ estimates derived from precipitation.

From the perspective of simulation accuracy, the $ET_{P\text{-}LSH\_\theta}$ driven by $\theta_{GLDAS\ Noah}$ performed best, outperforming estimates derived from any other soil moisture and precipitation (Table 3). The satellite-based $\theta_{ESA\ CCI}$ showed poor performance, which may be attributed to missing data, and simple temporal linear interpolation weakened the seasonal variation in soil moisture. The $ET_{P\text{-}LSH\_\theta}$ estimates derived from $\theta_{Qu}$ and $\theta_{Yang}$ performed well, where soil moisture came from machine learning and model assimilation, respectively. By contrast, the $ET_{P\text{-}LSH\_P}$ estimates overall had high and relatively stable precision, with RMSE of $7.70 \pm 0.46$ mm month$^{-1}$, while it was $8.39 \pm 1.08$ mm month$^{-1}$ from five $ET_{P\text{-}LSH\_\theta}$ estimates.

**Table 3.** RMSE (mm), Bias (mm), NSE, and R$^2$ of the five $ET_{P\text{-}LSH\_\theta}$ and five $ET_{P\text{-}LSH\_P}$ in comparison with the $ET_{recon}$ for aggregation of two basins.

| Soil moisture sources | $ET_{P\text{-}LSH\_\theta}$ | | | | Precipitation sources | $ET_{P\text{-}LSH\_P}$ | | | |
|---|---|---|---|---|---|---|---|---|---|
| | RMSE | Bias | NSE | R$^2$ | | RMSE | Bias | NSE | R$^2$ |
| $\theta_{Qu}$ | 7.57 | -2.88 | 0.82 | 0.86 | $P_{MSWEP}$ | 7.17 | -0.82 | 0.84 | 0.86 |
| $\theta_{ESA\ CCI}$ | 10.92 | -7.07 | 0.63 | 0.81 | $P_{GPM}$ | 7.76 | 2.15 | 0.81 | 0.87 |
| $\theta_{GLDAS\ Noah}$ | 6.44 | -1.97 | 0.87 | 0.89 | $P_{GLDAS\ Noah}$ | 8.22 | -0.22 | 0.79 | 0.81 |
| $\theta_{MERRA}$ | 9.64 | -7.09 | 0.71 | 0.87 | $P_{MERRA}$ | 8.05 | 1.86 | 0.80 | 0.86 |
| $\theta_{Yang}$ | 7.36 | -1.93 | 0.83 | 0.86 | $P_{CMFD}$ | 7.28 | -2.26 | 0.83 | 0.85 |



## 5 Discussion and conclusion

This study discussed the application of remote sensing ET algorithms to alpine barren areas on the TP. The first part of this study investigated the applicability of six existing coupling algorithms with $ET_{recon}$ in two basins. The moisture constraints and equilibrium equations for these algorithms were different. The A1 algorithm, which considers cumulative precipitation and equilibrium evaporation in soil evaporation, has the best performance on a monthly scale for both basins, with RMSE of 4.06 mm month$^{-1}$ for the Qaidam Basin and RMSE of 11.13 mm

month$^{-1}$ for the Qiangtang Plateau. The A5 algorithm, which directly considers soil moisture as a constraint, is second in performance, with RMSE of 4.36 mm month$^{-1}$ for the Qaidam Basin and RMSE of 11.42 mm month$^{-1}$ for the Qiangtang Plateau. The ET estimates from the A2 algorithm hardly match well for both basins because they are significantly affected by high-quality soil properties. The A4 algorithm uses the diurnal temperature range to reflect the apparent thermal inertia and humidity constraints, with moderate performance in both basins. Both algorithms,

A3 and A6, use an $RH^{VPD/k}$ term to express the sensitivity of the soil water deficit, and take the assumption that the surface moisture status is reflected in the adjacent atmospheric moisture, specifically in the form of evaporative demand of the atmosphere. This method has good applicability for ET estimation (Fisher et al., 2008; Zhang et al., 2010a; Mu et al., 2011), which may be because it pays more attention to total ET rather than soil evaporation. On the barrens of the TP, vegetation is sparse, and only soil evaporation exists; therefore, defects involving this method are

exposed. On the TP, RH has weak seasonality, whereas VPD is high in summer and low in winter, with strong seasonal variability. These phenomena result in $RH^{VPD/k}$ being high in winter and low in summer, which is contrary to actual soil moisture. In addition, the relationship between VPD and soil moisture may be decoupled on a daily scale (Purdy et al., 2018; Brust et al., 2021), which will eventually lead to model structural errors involving the A3 and A6 algorithms.

The second part of this study improved the P-LSH algorithm by introducing two frameworks for quantifying moisture constraints to ET in terms of surface soil moisture and precipitation. From the perspective of the regional average, the two improved algorithms significantly improved the performance of the P-LSH algorithm, and the simulation accuracy was higher than that of the six existing coupling algorithms. The P-LSH$_\theta$ algorithm showed the best performance among all algorithms, indicating that direct soil moisture can adequately express the moisture

supply in evaporation estimates for barrens. As a surrogate for moisture supply, precipitation can better express the constraints in barrens evaporation than RH, VPD, ATI, etc. However, the two estimates show some uncertainty in the Qiangtang Plateau, which requires more soil evaporation observations or other means to further estimate their reliability.

    The last part of this study tested the applicability of satellite soil moisture and precipitation data for improving ET

retrieval and analyzing the influence of soil moisture and precipitation uncertainties on ET estimation on alpine barren areas. In the spatial pattern, the uncertainty of $E_{s\_P\text{-}LSH\_\theta}$ was lower because the model structure flattened the magnitude difference in soil moisture. On the Qiangtang Plateau, the uncertainty of $E_{s\_P\text{-}LSH\_P}$ is larger, with 47.4% of the $Cv$ higher than 0.5, which is mainly due to the underestimation of precipitation by the GPM and MERRA in the northwestern basin. From the perspective of the regional average, the uncertainty of soil moisture is comparable

to that of precipitation, yet the uncertainty of $E_{s\_P\text{-}LSH\_\theta}$ is higher than that of $E_{s\_P\text{-}LSH\_P}$. The $ET_{P\text{-}LSH\_\theta}$ derived from





θ<sub>GLDAS Noah</sub> performs better than those from any other soil moisture and precipitation datasets, and the $ET_{P\text{-}LSH\_P}$ from all precipitation datasets generally showed high and stable accuracy, suggesting that high-quality soil moisture can optimally express moisture supply to ET, and that more accessible precipitation data can serve as a substitute of soil moisture as an indicator of moisture status for its robust performance in barren evaporation.

There were some uncertainties in this study. Because the revisit rates of various satellites are usually two to three days, it is difficult to obtain full daily soil moisture coverage of basins, and the satellite-based $\theta_{ESA\ CCI}$ faces the risk of spatial or temporal discontinuity. Simple temporal linear interpolation was used in our study, which weakened the seasonality of soil moisture. Although differences in various soil moisture datasets were discussed in this study, more spatially and temporally continuous satellite-based soil moisture datasets would be of significant interest.

Considering the coarse spatial resolution, uncertainties in the GRACE data are generally much greater; therefore, the $ET_{recon}$ estimates derived from it also have a coarse temporal and spatial resolution (monthly and basin-scale) and high uncertainty. We matched the pixel-scale and daily remote sensing algorithm outputs with the $ET_{recon}$, which may cause errors offset in the algorithms to a certain extent. In addition, various processes for GRACE products are sources of uncertainty in $\Delta S$, which in turn affects the accuracy of the $ET_{recon}$. Despite the above uncertainties, the

water balance method is still an effective means of providing a benchmark for remote sensing algorithm outputs at a basin-scale and is recognized in most studies (Zeng et al., 2012; Long et al., 2014). In terms of results, almost all algorithms had high uncertainty in the simulation of soil evaporation on the Qiangtang Plateau, especially in the summer of 2006 and subsequent years. Zhang et al. (2017) reported that inland lakes on the Qiangtang Plateau have expanded since the 1990s, whereas static land cover was used in this study. In the future, a dynamic dataset will be

necessary to reflect the characteristics of the ground surface for ET estimation.

**Code and data availability**

The code of the original and improved P-LSH algorithms used in this study are available from the corresponding author (kzhang@hhu.edu.cn). All data for this paper are properly cited and referred in Table 1. Specifically, the

meteorological data from CMFD are available at http://data.tpdc.ac.cn/; the radiation data from CERES SYN1deg are available at https://ceres.larc.nasa.gov/; the NDVI data from MODIS are available at https://lpdaac.usgs.gov; the soil moisture data from *The Soil Moisture Dataset of China Based on Microwave Data Assimilation* and *Land Surface Soil Moisture Dataset of SMAP Time-Expanded Daily 0.25°×0.25° over Qinghai-Tibet Plateau Area* are available at http://data.tpdc.ac.cn/; the soil moisture data from ESA CCI are available at https://esa-soilmoisture-

cci.org/; the precipitation data from GPM are available at https://gpm.nasa.gov/; the precipitation data from MSWEP are available at http://www.gloh2o.org/; GLDAS Noah data are available at https://disc.gsfc.nasa.gov/; MERRA data are available at https://gmao.gsfc.nasa.gov/; the land cover data from MCD12Q1 are available at https://lpdaac.usgs.gov/; the soil properties from *A China Dataset of Soil Hydraulic Parameters Pedotransfer Functions for Land Surface Modeling* are available at http://data.tpdc.ac.cn/; the reconstructed ET estimates of

Qaidam Basin and Qiangtang Plateau are available in Li et al. (2014).

**Author contribution**





JF and KZ conceived the idea and designed the research. JF and HZ performed the calculation. JF, KZ, HZ and LC conducted the analysis. All authors contributed to results discussion and manuscript writing.


**Competing interests**

The authors declare that they have no conflict of interest.

**Acknowledgments**

This study was supported by the National Key Research and Development Program of China (2018YFC1508101), National Natural Science Foundation of China (51879067), Natural Science Foundation of Jiangsu Province (BK20180022), and Six Talent Peaks Project in Jiangsu Province (NY-004). We gratefully acknowledge Professor Lei Wang from the Institute of Tibetan Plateau Research, Chinese Academy of Sciences for his help in providing the reconstructed ET estimates derived from the terrestrial water balance method used in this study.

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
