# Peer review of "Improved Soil Evaporation Remote Sensing Retrieval Algorithms and Associated Uncertainty Analysis on the Tibetan Plateau"

_Hydrology and Earth System Sciences, 2022_

## Author Comment (AC2)

**Response to Reviewers' comments**

We greatly appreciate the anonymous referee for providing valuable and constructive comments that are of great help for us to improve the quality of the manuscript. We have fully considered the comments and will revise the manuscript accordingly. The point-to-point responses to the comments and our plans for revision are listed below.

***Replies to the General Comments:***

1. *This manuscript improves satellite-based algorithm for estimating soil evaporation by adding the frameworks for quantifying moisture constraints to ET into P-LSH model, and assesses the impact of moisture constraint uncertainty on the estimated ET. Mechanism studies about ET and their components (e.g., transpiration, soil evaporation etc) in alpine barren areas, especially for Tibetan Plateau (TP), are very limited, and this study of ET mechanism in TP region is quite necessary. There are still some issues should be addressed before a publication.*

**Response:**

Thanks for your positive evaluation and encouraging comments on our manuscript. Our point-to-point responses to your comments are listed below.

2. *The authors have paid more attention to soil evaporation, and neglected the vegetation transpiration. For example, in line 538-539 of page 23, "On the barrens of the TP, vegetation is sparse, and only soil evaporation exists". This addressing is not very rigorous. Grasslands account for 20.2% in Qaidam basin and 39.7% in Qiangtang Plateau, respectively, and Nelson et al (2020) indicate that transpiration in grasslands accounts for 40%-60% ET during growing seasons. I think that the authors should pay some attentions to transpiration estimation by considering the uncertainties of some others vegetation canopy conductance models. And, it is not clear the canopy conductance is calculated by which model; is it the empirical relationship between conductance and climatic variables, or the Jarvis-Stewart model? If the later, Jarvis model has poor performances in capturing the responses of conductance to climatic variables (e.g. air temperature), compared to other models such as Ball-Berry model, Ball-Berry-Leuning model and Mdelyn model. The uncertainties caused by choice of conductance model on ET may result in 32%-53% errors (Zhao et al., 2020). Therefore, I suggest the authors can also consider the influences of vegetation conductance model on estimated ET in TP.*

**Response:**

Thanks for your comments and suggestions. There is a misunderstanding here. I indeed estimated transpiration in this study. However, the objective of this study is to improve the soil evaporation algorithm in the existing P-LSH algorithm. We have optimized our transpiration algorithm in another recent study. In this study, the estimation of ET is conducted on the pixel scale and then aggregated to a basin. For these vegetated pixels, ET is estimated by the existing optimized P-LSH algorithm.

The barren is a kind of non-vegetated land cover, so we reasonably think that only soil evaporation exists on barrens.

In grassland pixels, we took the revised P-LSH algorithm to estimate transpiration. The calculation of canopy conductance takes from an NDVI-based Jarvis–Stewart-type, with NDVI quantifying the biome-dependent potential canopy conductance and with temperature, VPD, $CO_2$, shortwave radiation, and soil moisture reducing to the actual value. The method was developed and validated with flux towers in Tibetan grassland in a previous manuscript, and the transpiration estimations from our revised canopy conductance framework show considerable agreement with observations from flux towers (see Figure R1). The manuscript describing the transpiration improvement was just accepted by Journal of Hydrology and will be published online soon. The authors understand and agree on the significance of canopy conductance and the associated uncertainty analysis, but it is not the key part of this study because the vegetation is not the major part of the basins that we selected.

[Figure]

**Figure R1.** Time series of daily measured and modelled evapotranspiration (AET: W m$^{-2}$) using the improved P-LSH algorithm driven by tower-measured and reanalysis meteorology for three grassland flux towers. The time series and statistics of modelled AET driven by tower-measured and reanalysis meteorology are marked in orange and brown, respectively.

3. *The method description for estimating soil evaporation is not clear in section 3. The authors introduced five existing soil evaporation algorithms and then proposed two improvements. In each algorithm, descriptions of main parameters are needed. For example, how the biome-specific constants are determined in the PM-Brust soil evaporation algorithm?*

**Response:**

Thanks for your suggestion. In six existing and two improved soil evaporation algorithms, there are parameter k in A3, parameters rbl$_{max}$, rbl$_{min}$, VPD$_{open}$, VPD$_{close}$ in A5, parameters k and r$_{tot}$ in A6, and parameter r$_{tot}$ in P-LSH$_{\theta}$ (in P-LSH$_{P}$ it takes the same value as that in P-LSH$_{\theta}$). Only parameters k and r$_{tot}$ in A6 are precalibrated values, other parameters lack prior values for the barren type. The parameter k in A3, parameter r$_{tot}$ in P-LSH$_{\theta}$, and parameters rbl$_{max}$, rbl$_{min}$, VPD$_{open}$, VPD$_{close}$ in A5 are calibrated. Actually, we listed the parameter values in the Results section (Lines 345, 350, 352, 367, and 404-406), and we will add more descriptions in the Methodology part in the revised manuscript.

*4. What is the difference between P-LSH soil evaporation algorithm (P-LSH$_p$) and PML soil evaporation algorithm? How the potential evaporation was calculated? Actually, it is not fair to compare soil evaporation algorithms with different potential evaporation equations. If the authors use the same equations, it is reasonable to compare soil evaporation algorithms. And, the difference between P-LSH soil evaporation algorithm (P-LSH$_\theta$) and PML soil evaporation algorithm is fwet. Why the authors do not add the fwet into P-LSH$_\theta$?*

**Response:**

Thanks for your question. The soil evaporation in P-LSH$_P$ takes a similar structure as that in PML, their sole discrepancy focuses on the estimation of the equilibrium (that is potential) evaporation, which both controls the soil evaporation and moisture constraint. The PML takes the simplified Priestley-Taylor-type equation as the equilibrium values, while P-LSH$_P$ takes the Penman-Monteith-type equation that considers the effect of the vapor pressure and resistance on equilibrium evaporation.

We conducted the comparison for the following reasons. Firstly, we think the two algorithms are comparable because both of them produce the actual values, and they are evaluated by a benchmark. Besides, we want to know with a similar structure, whether a theoretical better estimation of the equilibrium evaporation could contribute to the estimation of actual evaporation and moisture constraint, and it indeed turned out that way. By comparing the two, we can also separate the impact of potential evaporation on actual values.

For the latter question, we guess that there is a typo. The reviewer probably means that "And, the difference between P-LSH soil evaporation algorithm (P-LSH$_\theta$) and PM-Brust soil evaporation algorithm is fwet". Actually, the discrepancy between P-LSH$_\theta$ and PM-Brust is not only in f$_{wet}$ but also in the estimation of the terms $g_{a\_s}$ and $g_{totc}$ (a correction item of $r_{tot}$). In the P-LSH$_\theta$, the $r_{tot}$ is a sensitive parameter that is estimated by calibration, while in PM-Brust, the $r_{tot}$ is determined by VPD and four biome-specific constants. In the estimation of the term $g_{a\_s}$, the conductance to convective heat transfer ($g_{ch}$) is a biome-specific constant in the P-LSH$_\theta$, while in the PM-Brust, it is assumed to be equal to $r_{tot}$. We will make a clearer description in the revised manuscript.

In terms of f$_{wet}$, it is a term to divide the saturated surface and moist surface, and takes a value of 0 if the relative humidity is lower than 70% (Mu et al., 2011). It is always dry in the Tibetan Plateau, and about 97% of the pixels and days have a relative humidity below 70% in our study areas, this indicates the *f$_{wet}$* mostly does not impact the calculation. Nevertheless, for the integrity of the algorithm, we will add it to the revised manuscript.

*5. Figure 3 and 4 have showed the results of A1-A6 for five existing soil evaporation algorithms. I suggest that two improvement soil evaporation algorithms proposed by the authors should be added into the comparisons.*

**Response:**

Thanks. We will add them in the revised version.

***Replies to the Specific Comments:***

1. *Line 68: "32 days" is right?*

**Response:**

In Zhang et al. (2010), they summed precipitation and equilibrium evaporation over four periods prior and four periods after the current period to estimate $f$, with each period constituted of 8 days. Later they simplified the estimation and only considered the previous 32 days (in the supplementary information of Zhang et al. (2019)). Thanks for your suggestion. We will modify the description as follows: "Zhang et al. (2019) selected the cumulative precipitation and cumulative equilibrium evaporation rates over the past 32 days to estimate $f$, based on which a continuous ET dataset including each component was generated."

2. *Sometimes, the logical relationship between some context sentences is not strong. For example, line 110-111: "Saline lakes and deserts cover approximately one-quarter and one-third of the Qaidam Basin, respectively. This region is thus very dry.".*

**Response:**

Thanks for your suggestion, we will delete the sentence "*This region is thus very dry.*" and thoroughly check the logicality throughout the whole manuscript.

3. *Figure 1 should include scale bar and compass.*

**Response:**

Thanks for your suggestion, we will add them in the revised version.

4. *Line 301: the description "vegetation evapotranspiration" is not right.*

**Response:**

Thanks for your mention. Here we do not just refer to the transpiration process of vegetation, instead, it is the total evapotranspiration framework including canopy transpiration and soil evaporation in a vegetation pixel.

Reference:

Mu, Q., Zhao, M., and Running, S. W.: Improvements to a MODIS global terrestrial evapotranspiration algorithm, Remote Sensing of Environment, 115, 1781-1800, https://doi.org/10.1016/j.rse.2011.02.019, 2011.

Zhang, Y., Leuning, R., Hutley, L. B., Beringer, J., McHugh, I., and Walker, J. P.: Using long term water balances to parameterize surface conductances and calculate evaporation at 0.05° spatial resolution, Water Resources Research, 46, W05512, https://doi.org/10.1029/2009WR008716, 2010.

Zhang, Y., Kong, D., Gan, R., Chiew, F. H. S., McVicar, T. R., Zhang, Q., and Yang, Y.: Coupled estimation of 500 m and 8-day resolution global evapotranspiration and gross primary production in 2002-2017, Remote Sensing of Environment, 222, 165-182, https://doi.org/10.1016/j.rse.2018.12.031, 2019.

---

## Author Response (AR1)

**Revision Notes for Manuscript hess-2022-210**

Editor comments: I would like to thank your responses. Please upload a point-by-point response, a revised manuscript, and a version with tracked changes. Specifically, please try to clarify the novelty of this study in the abstract and introduction, explain the treatment of different ET components, and improve the presentation of different soil evaporation algorithms as well as their inter-comparisons.

**Response:**

We thank Editor for your valuable remarks. The novelty of this study has been clarified and reorganized in lines 15-25 and 105-108. In addition, the treatment of different ET components is explained in the response to the Reviewer #2's General Comments #2. Moreover, the presentation of different soil evaporation algorithms is improved in the revised manuscript (see lines 265-267, 280-281, 288-289, 306-309, 379-382, and 396-400). Moreover, we have thoroughly revised our manuscript based on the reviewer comments. The revised manuscript has been largely improved. Detailed point-to-point responses are provided below.

**Response to Reviewer#1's Comments**

**Please refer to the "changes\_tracked" version of our manuscript, which is attached by the end of the Notes, to see our detailed edits and revisions.**

**Replies to the General Comments:**

This manuscript tried to improve the satellite-based land surface ET algorithm by introducing soil moisture. Using satellite data to simulate the water cycle or calibrate models are very attractive considering the growing availability of remote sensing data. The method is reasonable, the findings are useful, and the article was well written. However, there are still some minor issues that need to be addressed.

**Response:**

Thanks for your positive evaluation and encouraging comments on our manuscript. Our point-to-point responses to your comments are listed below.

**Replies to the Specific Comments:**

1. *L15: '... introducing two frameworks ...' there are no any information about these 'two' in this sentence. It would be better combine this and the following sentence.* **Response:**

Thanks for your suggestion. We have modified it as follows: "In this study, we aimed to improve the satellite-driven Process-based Land Surface ET/Heat fluxes algorithm (P-LSH) for better satellite retrieval of ET on the Tibetan Plateau by introducing two effective soil moisture constraint schemes, in which normalized surface soil moisture and the ratio of cumulative antecedent precipitation to cumulative antecedent equilibrium evaporation are used to represent soil water stress, respectively, based on the intercomparison and knowledge-learning of the existing schemes. We first conducted intercomparison of six existing soil evaporation algorithms and sorted out the two most effective soil moisture constraint schemes. We then introduced the modified versions of the two constraint schemes into the P-LSH algorithm and further optimized the parameters using the Differential Evolution method. As a result, it formed two improved P-LSH algorithms." (see lines 15-25).

**2. *L20: 'two improved P-LSH algorithm' seems not clear. What are they?* **Response:**

Thanks for your suggestion. Because we introduced two schemes to quantifying moisture constraints to ET in the original P-LSH algorithm, it correspondingly leads to two improved P-LSH algorithms. To avoid any confusion, we revised the description to make the sentence clearer, as shown in *Comment* #1 and lines 15-25.

**3. *It is better to highlight the significant point of the study further.* **Response:**

Thanks for your suggestion. We conducted this study based on the following background. On one hand, the Tibetan Plateau is crucial for Asian monsoon development and regional to global water and energy cycles, but relatively few studies have been carried out on its barren areas because of its remoteness and limited ground data. Remote sensing retrieval can conveniently estimate ET in this region, but their accuracy needs to be further assessed and improved. On the other hand, some studies (Zhang et al., 2015; Pan et al., 2020) have shown that water supply/soil moisture constraint is one of the key controlling factors of ET in arid and semi-arid regions, whose mathematical representation are rarely systematically assessed or discussed in existing studies.

In this paper, the applicability and effectiveness of various moisture constraint schemes in the existing ET algorithms in typical arid/semi-arid basins were first analysed. Based on the above analysis, we then proposed two improved P-LSH algorithms, in which normalized surface soil moisture and the ratio of cumulative antecedent precipitation to cumulative antecedent equilibrium evaporation are used to represent soil water stress, respectively. We further assessed the impacts of uncertainty in the soil moisture and precipitation forcing data on the soil evaporation retrievals. Therefore, we further explicitly stated the contributions of this study and highlighted the significant points of our study in the introduction and conclusion sections (see lines 105-108 and 556-561).

4. *Table 1: why use 30''* **Response:**

As we mentioned in the caption of Table 1, we listed the original resolutions of the datasets. The soil properties dataset is in a raster format with a resolution of 30 arc seconds. To match with the spatial resolutions of the other inputs in the ET algorithm, we aggregated the dataset from the original 30" resolution to  $1/12^{\circ}$  using the arithmetic averaging method.

**5. Figure 3 and 4: More information (A1, ...; $ET_{recon}$ ) is needed in the caption to make it be understandable.**

**Response:**

Thanks for your suggestion. The A1, A2, A3, A4, A5, A6 are the combined ET algorithms that combine the vegetation evapotranspiration scheme and water evaporation scheme in the original P-LSH algorithm with one of the six existing soil evaporation algorithms (see Table 2). The  $ET_{recon}$  item represents the reconstructed ET estimates derived from the terrestrial water balance method. We have added more information in the caption of Figures 3 and 4 (see lines 380-382 and 397-399).

**6. *I am not quite clear how the authors evaluate the 'uncertainty'*. **Response:**

In the improved algorithms, the precipitation and soil moisture data are used to express the moisture constraint on ET. We investigated the impact of various precipitation and soil moisture datasets on the ET to quantify the impacts of uncertainty in the key inputs on model outputs. Taking the P-LSH $\theta$  algorithm as an example, we investigated the variation between multiple ET estimates derived from multiple soil moisture datasets, and as a comparison, we also investigated the variation of multiple soil moisture datasets. The same method is also applied to the uncertainty evaluation of P-LSHP algorithm, as shown in Figures 10-12. By quantifying the variations between soil moisture/precipitation forcing data and their corresponding ET estimates, the characteristics and uncertainties of the improved algorithms are discussed in Section 4.3. To make it clearer, we have provided more description in the revised manuscript (see lines 477-485).

**Reference:**

Pan, S., Pan, N., Tian, H., Friedlingstein, P., Sitch, S., Shi, H., Arora, V. K., Haverd, V., Jain, A. K., Kato, E., Lienert, S., Lombardozzi, D., Nabel, J. E. M. S., Ottlé, C., Poulter, B., Zaehle, S., and Running, S. W.: Evaluation of global terrestrial evapotranspiration using state-of-theart approaches in remote sensing, machine learning and land surface modeling, Hydrology and Earth System Sciences, 24, 1485-1509, https://doi.org/10.5194/hess-24-1485-2020, 2020.

Zhang, K., Kimball, J. S., Nemani, R. R., Running, S. W., Hong, Y., Gourley, J. J., and Yu, Z.: Vegetation greening and climate change promote multidecadal rises of global land evapotranspiration, Scientific Reports, 5, 15956, https://doi.org/10.1038/srep15956, 2015.

**Response to Reviewer#2's Comments**

**Please refer to the "changes\_tracked" version of our manuscript, which is attached by the end of the Notes, to see our detailed edits and revisions.**

**Replies to the General Comments:**

1. This manuscript improves satellite-based algorithm for estimating soil evaporation by adding the frameworks for quantifying moisture constraints to ET into P-LSH model, and assesses the impact of moisture constraint uncertainty on the estimated ET. Mechanism studies about ET and their components (e.g., transpiration, soil evaporation etc) in alpine barren areas, especially for Tibetan Plateau (TP), are very limited, and this study of ET mechanism in TP region is quite necessary. There are still some issues should be addressed before a publication.

**Response:**

Thanks for your positive evaluation and encouraging comments on our manuscript. Our point-to-point responses to your comments are listed below.

2. The authors have paid more attention to soil evaporation, and neglected the vegetation transpiration. For example, in line 538-539 of page 23, "On the barrens of the TP, vegetation is sparse, and only soil evaporation exists". This addressing is not very rigorous. Grasslands account for 20.2% in Qaidam basin and 39.7% in Qiangtang Plateau, respectively, and Nelson et al (2020) indicate that transpiration in grasslands accounts for 40%-60% ET during growing seasons. I think that the authors should pay some attentions to transpiration estimation by considering the uncertainties of some others vegetation canopy conductance models. And, it is not clear the canopy conductance is calculated by which model; is it the empirical relationship between conductance and climatic variables, or the Jarvis-Stewart model? If the later, Jarvis model has poor performances in capturing the responses of conductance to climatic variables (e.g. air temperature), compared to other models such as Ball-Berry model, Ball-Berry-Leuning model and Mdelyn model. The uncertainties caused by choice of conductance model on ET may result in 32%-53% errors (Zhao et al., 2020). Therefore, I suggest the authors can also consider the influences of vegetation conductance model on estimated ET in TP.

**Response:**

Thanks for your comments and suggestions. There is a misunderstanding here. We indeed estimated transpiration in this study. However, the objective of this study is to improve the soil evaporation algorithm in the existing P-LSH algorithm. We have optimized our transpiration algorithm in another recent study (see Feng et al. (2022)). In this study, the estimation of ET is conducted on the pixel scale and then aggregated to the basin level (see Figure 2). For these vegetated pixels, ET is estimated by the existing optimized P-LSH algorithm with an optimized transpiration scheme published in Feng et al. (2022). The barren is a kind of non-vegetated land cover, so we reasonably think that only soil evaporation exists on barren pixels.

In grassland pixels, we took the existing optimized P-LSH algorithm to estimate transpiration. The calculation of canopy conductance takes from an NDVI-based Jarvis–Stewart-type, with NDVI quantifying the biome-dependent potential canopy conductance and with air temperature, VPD, CO2, shortwave radiation, and soil moisture serving as environmental stress factors. The method was developed and validated with the observations of flux towers in the Tibetan grasslands in Feng et al. (2022). The transpiration estimations from our revised canopy conductance framework show considerable agreement with observations from flux towers (see Figure R1). The manuscript describing the transpiration improvement was just published (Feng et al., 2022). The authors understand and agree on the significance of canopy conductance and the associated uncertainty analysis, but it is not the key part of this study because the vegetation is not the major part of the basins that we selected.

**Figure R1.** Time series of daily measured and modelled evapotranspiration (AET: W m-2) using the improved P-LSH algorithm driven by tower-measured and reanalysis meteorology for three grassland flux towers. The time series and statistics of modelled AET driven by tower-measured and reanalysis meteorology are marked in orange and brown, respectively.

3. The method description for estimating soil evaporation is not clear in section 3. The authors introduced five existing soil evaporation algorithms and then proposed two improvements. In each algorithm, descriptions of main parameters are needed. For example, how the biome-specific constants are determined in the PM-Brust soil evaporation algorithm?

**Response:**

Thanks for your suggestion. In the six existing and two improved soil evaporation algorithms, there are parameter k in A3, parameters  $rbl_{max}$ ,  $rbl_{min}$ ,  $VPD_{open}$ , and  $VPD_{close}$  in A5, parameters k and  $r_{tot}$  in A6, and parameter  $r_{tot}$  in P-LSH $\theta$  (P-LSHP share the same parameter values as P-LSH $\theta$ ). Only parameters k and  $r_{tot}$  in A6 are set to the precalibrated values, while the other parameters lack prior values for the barren type. The parameter k in A3, parameter  $r_{tot}$  in P-LSH $\theta$ , and parameters  $rbl_{max}$ ,  $rbl_{min}$ ,  $VPD_{open}$ , and  $VPD_{close}$  in A5 are calibrated. Actually, we listed the parameter values in the Results section in the original manuscript and are further updated in the revised manuscript (see lines 366-367, 370-372, 373-374, 390-391, and 428-432). We have further provided more descriptions on these parameters and their values in the Methodology section (see lines 265-267, 299-302 and 308-309).

4. What is the difference between P-LSH soil evaporation algorithm (P-LSHp) and PML soil evaporation algorithm? How the potential evaporation was calculated? Actually, it is not fair to compare soil evaporation algorithms with different potential evaporation equations. If the authors use the same equations, it is reasonable to compare soil evaporation algorithms. And, the difference between P-LSH soil evaporation algorithm (P-LSH $\theta$ ) and PML soil evaporation algorithm is fwet. Why the authors do not add the fwet into P-LSH $\theta$ ?

**Response:**

Thanks for your question. The soil evaporation algorithm of  $P-LSH_P$  takes a similar structure as that of PML. However, the only difference in the two algorithms lies in the estimation of the equilibrium (potential) evaporation, which regulates soil evaporation and moisture constraint. The PML soil evaporation algorithm takes the simplified Priestley-Taylor-type equation to estimate the equilibrium values, while  $P-LSH_P$  takes the Penman-Monteith-type equation, which considers the impacts of the vapor pressure and resistance on the equilibrium evaporation.

We conducted the comparison for the following reasons. Firstly, we think the two algorithms are comparable because both of them produce the actual values, and they are assessed by the same benchmark data. Besides, we want to tell whether a theoretical better estimation of the equilibrium evaporation within a similar soil evaporation estimation framework can contribute to the estimation of actual evaporation and moisture constraint. Our results indeed show that the estimation of equilibrium evaporation matters for estimating soil evaporation in these methods. By comparing the two methods, we can also separate the impact of potential evaporation on actual values.

For the latter question, we guess that there is a typo. The reviewer probably means that "And, the difference between P-LSH soil evaporation algorithm (P-LSH $\theta$ ) and PM-Brust soil evaporation algorithm is fwet". Actually, the difference between P-LSH $\theta$  and PM-Brust is not only in fwet but also in the estimation of the terms ga\_s and gtote (a correction item of rtot). In the P-LSH $\theta$ , the rtot is a sensitive parameter that is estimated through calibration, while in PM-Brust, the rtot is determined by VPD and four biome-specific constants. In the estimation of the term ga\_s, the conductance to convective heat transfer (gch) is a biome-specific constant in the P-LSH $\theta$ , while in the PM-Brust, it is assumed to be equal to rtot. We have made a clearer description in lines 288-289 and 294-299.

In terms of  $f_{wet}$ , it is a term to divide the saturated surface and moist surface, and takes a value of 0 if the relative humidity is lower than 70% (Mu et al., 2011). It is always dry in the Tibetan Plateau, and relative humidity is below 70% in over 97% of the pixels and during 97% of the time in our study areas; this indicates the  $f_{wet}$  mostly does not impact the calculation. Nevertheless, for the integrity of the algorithm, we have added this term back to the algorithm and re-calculated the results (see line 294). It turns out that no obvious difference appears in the new calculations

5. Figure 3 and 4 have showed the results of A1-A6 for five existing soil evaporation algorithms. I suggest that two improvement soil evaporation algorithms proposed by the authors should be added into the comparisons.

**Response:**

Thanks. We have added them in the revised version (see lines 379 and 396).

**Replies to the Specific Comments:**

**1. Line 68: "32 days" is right? Response:**

In the original study of Zhang et al. (2010), they summed precipitation and equilibrium evaporation over four periods prior and after the current period to estimate f, with each period constituted of 8 days (namely, 64 days as a total). Later, they simplified the estimation and only considered the previous 32 days (in the supplementary information of Zhang et al. (2019)). Thanks for your suggestion. We have modified the description as follows: "Zhang et al. (2019) selected the cumulative precipitation and cumulative equilibrium evaporation rates over the past 32 days to estimate f, based on which a continuous ET dataset including each component was generated" (see lines 72-75).

 Sometimes, the logical relationship between some context sentences is not strong. For example, line 110-111: "Saline lakes and deserts cover approximately one-quarter and one-third of the Qaidam Basin, respectively. This region is thus very dry". Response:

Thanks for your suggestion, we have deleted the sentence "*This region is thus very dry*." and thoroughly checked the logicality throughout the whole manuscript.

**3. *Figure 1 should include scale bar and compass.* **Response:**

Thanks for your suggestion. We have added them in the revised version (see line 121).

**4. *Line 301: the description "vegetation evapotranspiration" is not right.* **Response:**

Thanks for pointing this out. Here we do not only refer it to the transpiration process of vegetation. Instead, it is the total evapotranspiration that includes both vegetation transpiration and evaporation of vegetation surface in a vegetation pixel. We further explained it as this term first appears (see line 314).

Reference:

Feng, J., Zhang, K., Chao, L., and Liu, L.: An improved process-based evapotranspiration/heat fluxes remote sensing algorithm based on the Bayesian and Sobol' uncertainty analysis framework using eddy covariance observations of Tibetan grasslands, Journal of Hydrology, 613, 128384, https://doi.org/10.1016/j.jhydrol.2022.128384, 2022.

Mu, Q., Zhao, M., and Running, S. W.: Improvements to a MODIS global terrestrial evapotranspiration algorithm, Remote Sensing of Environment, 115, 1781-1800, https://doi.org/10.1016/j.rse.2011.02.019, 2011.

Pan, S., Pan, N., Tian, H., Friedlingstein, P., Sitch, S., Shi, H., Arora, V. K., Haverd, V., Jain, A. K., Kato, E., Lienert, S., Lombardozzi, D., Nabel, J. E. M. S., Ottlé, C., Poulter, B., Zaehle, S., and Running, S. W.: Evaluation of global terrestrial evapotranspiration using state-of-theart approaches in remote sensing, machine learning and land surface modeling, Hydrology and Earth System Sciences, 24, 1485-1509, https://doi.org/10.5194/hess-24-1485-2020, 2020. Zhang, K., Kimball, J. S., Nemani, R. R., Running, S. W., Hong, Y., Gourley, J. J., and Yu, Z.: Vegetation greening and climate change promote multidecadal rises of global land evapotranspiration, Scientific Reports, 5, 15956, https://doi.org/10.1038/srep15956, 2015.

Zhang, Y., Leuning, R., Hutley, L. B., Beringer, J., McHugh, I., and Walker, J. P.: Using long - term water balances to parameterize surface conductances and calculate evaporation at 0.05 ° spatial resolution, Water Resources Research, 46, W05512, https://doi.org/10.1029/2009WR008716, 2010.

Zhang, Y., Kong, D., Gan, R., Chiew, F. H. S., McVicar, T. R., Zhang, Q., and Yang, Y.: Coupled estimation of 500 m and 8-day resolution global evapotranspiration and gross primary production in 2002-2017, Remote Sensing of Environment, 222, 165-182, https://doi.org/10.1016/j.rse.2018.12.031, 2019.